# Revealing nonlinear neural decoding by analyzing choices

Qianli Yang [1,6,7], Edgar Walker[2,3], R. James Cotton [4,5], Andreas S. Tolias [1,2,3] & Xaq Pitkow [1,2,3✉]

Sensory data about most natural task-relevant variables are entangled with task-irrelevant nuisance variables. The neurons that encode these relevant signals typically constitute a nonlinear population code. Here we present a theoretical framework for quantifying how the brain uses or decodes its nonlinear information. Our theory obeys fundamental mathematical limitations on information content inherited from the sensory periphery, describing redundant codes when there are many more cortical neurons than primary sensory neurons. The theory predicts that if the brain uses its nonlinear population codes optimally, then more informative patterns should be more correlated with choices. More specifically, the theory predicts a simple, easily computed quantitative relationship between fluctuating neural activity and behavioral choices that reveals the decoding efficiency. This relationship holds for optimal feedforward networks of modest complexity, when experiments are performed under natural nuisance variation. We analyze recordings from primary visual cortex of monkeys discriminating the distribution from which oriented stimuli were drawn, and find these data are consistent with the hypothesis of near-optimal nonlinear decoding.

[1] Department of Electrical and Computer Engineering, Rice University, Houston, TX, USA. [2] Department of Neuroscience, Baylor College of Medicine, Houston, TX, USA. [3] Baylor College of Medicine, Center for Neuroscience and Artificial Intelligence, Houston, TX, USA. [4] Shirley Ryan Ability Lab, Chicago, IL, USA. [5] Department of Physical Medicine and Rehabilitation, Northwestern University, Evanston, IL, USA. [6] Changzhou University, Aliyun School of Big Data, Changzhou, China. [7] Institute of Neuroscience, Key Laboratory of Primate Neurobiology, CAS Center for Excellence in Brain Science and Intelligence Technology, Chinese Academy of Sciences, Beijing, China. ✉email: xaq@rice.edu

How does an animal use, or 'decode', the information represented in its brain? When the average responses of some neurons are well-tuned to a stimulus of interest, this can be straightforward. In binary discrimination tasks, for example, a choice can be reached simply by a linear weighted sum of these tuned neural responses. Yet real neurons are rarely tuned to precisely one variable: variation in multiple stimulus dimensions influence their responses in complex ways. As we show below, when these nuisance variations have nonlinear effects on responses, they can dilute or even abolish the mean tuning to the relevant stimulus. Then the brain cannot simply use linear computation, nor can we understand neural processing using linear models.

A quantitative account of nonlinear neural decoding of sensory stimuli must first express how populations of neurons encode or represent information. Past theories of nonlinear population codes made unsupported assumptions about the covariability of this population responses[1,2], leading to substantially underestimated redundancy of large cortical populations. Here we correct this problem by generalizing information-limiting correlations[3] to nonlinear population codes, providing a more realistic theory of how much sensory information is encoded in the brain.

Just because a neural population encodes information, it does not mean that the brain decodes it all. Here, *encoding* specifies how the neural responses relate to the stimulus input, whereas *decoding* specifies how the neural responses relate to the behavioral output. To understand the brain's computational strategy we must understand how encoding and decoding are related, i.e. how the brain uses the information it has. These are distinct processes, so the brain could encode a stimulus well while decoding it poorly, or vice-versa.

This paper makes four main contributions. First, it weaves together important concepts about tuning curves and nuisance variables, nonlinear computation, and redundant population codes, forming a general, unified description of feedforward encoding and decoding processes in the brain. This description is supported by intuitive explanations and concrete examples to illustrate how these concepts relate to each other and enrich familiar views of neural computation. Second, this paper provides a simple way of testing the hypothesis that the brain's decoding strategy is efficient, using a simple statistic to assess whether neural response patterns that are informative about the task-relevant sensory input are also informative about the animal's behavior in the task. Third, it establishes the technical details needed to apply this test in practical neuroscience experiments. Fourth, we apply this test to analyze V1 data from macaque monkeys, finding direct experimental evidence for optimal nonlinear decoding.

The "Results" section describes the main concepts, their formal connections, and applications. More specifically, the first sections introduce a framework for understanding nonlinear computation, including basic notation, internal and external (nuisance) noise and their effects on information content and formatting, and how this information can be isolated by nonlinear computation of the right statistics. Subsequent sections introduce a formalism for decoding, including notions of linear and nonlinear choice correlations, fine and coarse estimation tasks, and predictions about those correlations under optimal decoding. This section continues by describing how redundancy in the population responses appears as special high-order response statistics, and how they affect the predictions. The last sections present an experimental application of these ideas. A sketch of the details of our general predictions are presented in the "Methods" section, and are derived in full in the Supplement along with details of their application to specific models and our experimental data.

## Results

**A simple example of a nonlinear code.** Imagine a simplified model of a visual neuron that includes an oriented edge-detecting linear filter followed by additive noise, with a Gabor receptive field like simple cells in primary visual cortex (Fig. 1a). If an edge is presented to this model neuron, different rotation angles will change the overlap, producing a different mean. This neuron is then tuned to orientation.

However, when the edge has the opposite polarity, with black and white reversed, then the linear response is reversed also. If the two polarities occur with equal frequency, then the positive and negative responses cancel on average. The mean response of this linear neuron to any given orientation is therefore precisely constant, so the model neuron is untuned.

Notice that stimuli aligned with the neuron's preferred orientation will generally elicit the highest or lowest response magnitude, depending on polarity. Edges evoking the largest response to one polarity will also evoke the smallest response to its inverse. Thus, even though the mean response of this linear neuron is zero, independent of orientation, the *variance* is tuned.

To estimate the variance, and thereby the orientation itself, the brain can compute the square of the linear responses. This would allow the brain to estimate the orientation independently from polarity. This is consistent with the well-known energy model of complex cells in the primary visual cortex, which use squaring nonlinearities to achieve invariance to the polarity of an edge[4].

Generalizing from this example, we identify edge polarity as a 'nuisance variable'—a property in the world that alters how task-relevant stimuli appear but is, itself, irrelevant for the current task (here, perceiving orientation). Other examples of nuisance

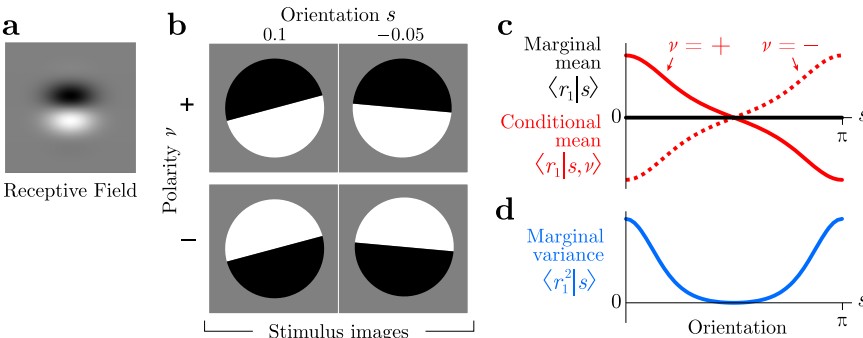

**Fig. 1 Simple nonlinear code for orientation induced by two polarities. a** Receptive field for a linear neuron. **b** Four example images, each with an orientation $s \in [0, \pi)$ and a polarity $\nu \in \{-1, +1\}$. **c** The mean response of the linear neuron is tuned to orientation if polarity were specified (conditional mean, red). But when the polarity is unknown and could take either value, the mean response is untuned (marginal mean, black). **d** Tuning is recovered by the marginal variance even if the polarity is unknown (blue).

variables include the illuminant for guessing a surface color, position for object recognition, the expression for face identification, or pitch for speech recognition. Generically, nuisance variables make it hard to extract the task-relevant variables from sense data, which is the central task of perception[5–10]. For example, cells in the early visual cortex are not tuned to object identity, since the object could appear at any location and V1 has not yet extracted the complex combinations of features that reveal object type independent of the nuisance variable of position. The brain learns from its history of sensory inputs which statistics of its many sense-data are tuned to the task-relevant variable. Good nonlinear computations then compute those statistics. In the orientation estimation task above, the relevant statistic was not the mean, but the variance.

**Task, stimuli, neural responses, actions**. Our mathematical framework describes a perceptual task, a stimulus with both relevant and irrelevant variables, neural responses, and behavioral choices.

In our task, an agent observes a multidimensional stimulus $(s, v)$ and must act upon one particular relevant aspect of that stimulus, $s$, while ignoring the rest, $v$. The irrelevant stimulus aspects serve as nuisance variables for the task ($v$ is the Greek letter 'nu' and here stands for nuisance). Together, these stimulus properties determine a complete sensory input that drives some responses $\mathbf{r}$ in a population of $N$ neurons according to the distribution $p(\mathbf{r}|s, v)$.

We consider a feedforward processing chain for the brain, in which the neural responses $\mathbf{r}$ are nonlinearly transformed downstream into other neural responses $\mathbf{R}(\mathbf{r})$, which in turn are used to create a perceptual estimate of the relevant stimulus $\hat{s}$:

$$(s, \nu) \rightarrow \mathbf{r} \rightarrow \mathbf{R} \rightarrow \hat{s} \tag{1}$$

We model the brain's estimate as a linear function of the downstream responses $\mathbf{R}$. Ultimately these estimates are used to generate an action that the experimenter can observe. We assume that we have recorded activity only from some of the upstream neurons, so we do not have direct access to $\mathbf{R}$, only a subset of $\mathbf{r}$. Nonetheless, we would like to learn something about the downstream computations used in decoding. In this paper, we show how to use the statistics of fluctuations in $\mathbf{r}$, $s$, and $\hat{s}$ to estimate the quality of nonlinear decoding.

We first develop the theory for local or fine-scale estimation tasks: the subject must directly report its estimate $\hat{s}$ for the relevant stimuli near a reference $s_0$, and we measure performance by the variance of this estimate, $\sigma_{\hat{s}}^2$. In later sections, we then generalize the problem to allow for binary discrimination as well as coarse tasks, which are more complicated mathematically but not conceptually different.

**Signal and noise**. The population response, which we take here to be the spike counts of each neuron in a specified time window, reflects both *signal* and *noise*, where the signal is the repeatable stimulus-dependent aspects of the response, and noise reflects trial-to-trial variation. Conventionally in neuroscience, the signal is often thought to be the stimulus dependence of the *average* response, i.e. the tuning curve $\mathbf{f}(s) = \sum_{\mathbf{r}} \mathbf{r}\, p(\mathbf{r}|s) = \langle \mathbf{r}|s \rangle$. (Angle brackets denote an average overall responses given the condition after the vertical bar.) Below we will broaden this conventional definition to allow the signal to include any stimulus-dependent statistical property of the population response.

Noise is the non-repeatable part of the response, characterized by the variation of responses to a fixed stimulus. It is convenient to distinguish *internal* noise from *external* noise. Internal noise is internal to the animal and is described by response distribution

$p(\mathbf{r}|s, v)$ when everything about the stimulus is fixed. This could also include uncontrolled variation in internal states[11–14], like attention, motivation, or wandering thoughts. External 'noise' is variability generated by the external world—nuisance variables—leading to a neural response distribution $p(\mathbf{r}|s)$ where only the relevant variables are held fixed. Both types of noise can lead to uncertainty about the true stimulus.

Trial-to-trial variability can of course be correlated across neurons. Neuroscientists often measure two types of second-order correlations: signal correlations and noise correlations[2,15–22]. Signal correlations measure shared variation in mean responses $\mathbf{f}(s)$ averaged over the set of stimuli $s$: $\rho_{\text{signal}} = \text{Corr}(\mathbf{f}(s))$ where again all averages are taken over all variables not fixed by a condition to the right of the vertical bar. (Internal) noise correlations measure shared variation that persists even when the stimulus is completely identical, nuisance variables and all: $\rho_{\text{noise}}(s, v) = \text{Corr}(\mathbf{r}|s, v)$.

For multidimensional stimuli, however, these correlations are only two extremes on a spectrum, depending on how many stimulus aspects are fixed across the trials to be averaged. We propose an intermediate type of correlation: *nuisance correlations*. Here we fix the task-relevant stimulus variable(s) $s$, and average over the nuisance variables $v$: $\rho_{\text{nuisance}}(s) = \text{Corr}(\mathbf{f}(s, v)|s)$. Including both internal and external (nuisance) noise correlations gives $\text{Corr}(\mathbf{r}|s)$.

Critically, but confusingly, some so-called 'noise' correlations and nuisance correlations actually serve as signals. This happens whenever the statistical pattern of trial-by-trial fluctuations depends on the stimulus, and thus contain information. For example, a stimulus-dependent noise covariance functions as a signal. There would still be true noise, i.e. irrelevant trial-to-trial variability that makes the signal uncertain, but it would be relegated to higher-order fluctuations[23] such as the variance of the response covariance (Fig. 2d, Table 1). Whether from internal or external noise, stimulus-dependent correlations lead naturally to nonlinear population codes, as we explain below.

**Nonlinear encoding by neural populations**. Most accounts of neural population codes actually address *linear* codes, in which the mean response is tuned to the variable of interest and completely captures all signals about it[3,24–27]. We call these codes linear because the neural response property needed to best estimate the stimulus near a reference (or even infer the entire likelihood of the stimulus, Supplement S.1.2.2) is a linear function of the response. Linear codes for different variables may arise early in sensory processing, like orientation in V1, or after many stages of computation[5,9], like for objects in the inferotemporal cortex.

If any of the relevant signals can only be extracted from nonlinear statistics of the neural responses[1,2], then we say that the population code is nonlinear (Table 1). One straightforward example is a stimulus-dependent covariance $Q(s) = \langle \mathbf{r}\mathbf{r}^{\top}|s \rangle$; its information can be decoded by quadratic operations $\mathbf{R} = \mathbf{r}\mathbf{r}^{\top}$ [28–30].

A simple example of a nonlinear code is the exclusive-or (XOR) problem. Given the responses of two binary neurons, $r_1$ and $r_2$, we would like to decode the value of a task-relevant signal $s = \text{XOR}(r_1, r_2)$ (Fig. 2a). We do not care about the specific value of $r_1$ by itself, and in fact, $r_1$ alone tells us nothing about $s$. The same is true for $r_2$. The usual view on nonlinear computation is that the desired signal can be extracted by applying an XOR or product nonlinearity. However, there is an underlying statistical reason this works: the signal is actually reflected in the trial-by-trial *correlation* between $r_1$ and $r_2$: when they are the same then $s = -1$, and when they are opposite then $s = +1$. The correlation,

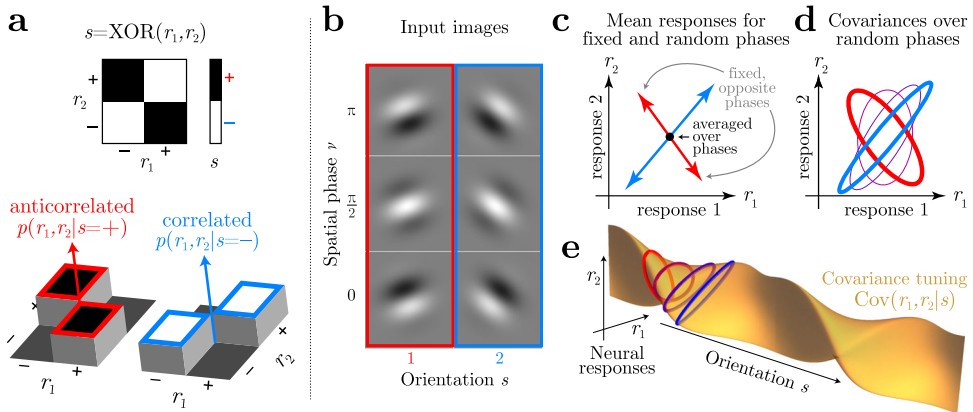

**Fig. 2 Nonlinear codes. a** Simple example in which a stimulus $s$ is the XOR of two neural responses (top). Conditional probabilities $p(r_1, r_2|s)$ of those responses (bottom) show they are anti-correlated when $s = +1$ (red) and positively correlated when $s = -1$ (blue). This stimulus-dependent correlation between responses creates a nonlinear code. The remaining panels show that a similar stimulus-dependent correlation emerges in orientation discrimination with an unknown spatial phase. **b** Gabor images with two orientations and three spatial phases. **c** Mean responses of linear neurons with Gabor receptive fields are sensitive to orientation when the phase is fixed (arrows), but point in different directions for different spatial phases. When phase is an unknown nuisance variable, this mean tuning, therefore, vanishes (black dot). **d** The response covariance $\text{Cov}(r_1, r_2|s)$ between these linear neurons is tuned to orientation even when averaging over spatial phase. Response covariances for four orientations are depicted by ellipses. **e** A continuous view of the covariance tuning to orientation for a pair of neurons.

**Table 1 Neural response properties relevant for linear and nonlinear codes.**

|  | Linear | Nonlinear | Quadratic |
|---|---|---|---|
| Trial data | $\mathbf{r}$ | $\mathbf{R(r)}$ | $\mathbf{rr}^\top$ |
| Signal | Mean($\mathbf{r}|s$) | Mean($\mathbf{R}|s$) | Mean($\mathbf{rr}^\top|s$) |
| Noise | Cov($\mathbf{r}|s$) | Cov($\mathbf{R}|s$) | Cov($\mathbf{rr}^\top|s$) |

In each case, the brain must estimate the stimulus from a single example of neural data, but the relevant function of that data is linear for linear codes and nonlinear for nonlinear codes (such as the quadratic example in the last column). The noise and signal can be quantified by the corresponding covariance and stimulus-dependent changes in the corresponding means (i.e. the tuning curve slope).

and thus the relevant variable $s$, can be estimated nonlinearly from $r_1$ and $r_2$ as $\hat{s} = -r_1 r_2$.

Some experiments have reported stimulus-dependent internal noise correlations that depend on the signal, even for a completely fixed stimulus without any nuisance variation[31–35]. Other experiments have turned up evidence for nonlinear population codes by characterizing the nonlinear selectivity directly[36–38].

More typically, however, stimulus-dependent correlations arise from external noise, leading to what we call nuisance correlations. In the introduction (Fig. 1) we showed a simple orientation estimation example in which fluctuations of an unknown polarity eliminate the orientation tuning of mean responses, relegating the tuning to variances. Figure 2b–e shows a slightly more sophisticated version of this example, where instead of two image polarities, we introduce spatial phase as a continuous nuisance variable. This again eliminates mean tuning but introduces nuisance covariances that are orientation tuned.

One might object that although the nuisance covariance is tuned to orientation, a subject cannot compute the covariance (or any other statistic of the encoding model) on a single trial because it does not experience all possible nuisance variables to average over. However, in linear codes, the subject does not have access to the tuned mean response $\langle \mathbf{r}|s \rangle$ either, just a noisy single-trial version of the mean, namely $\mathbf{r}$. Analogously, the subject does not need access to the tuned covariance, just a noisy single-trial version of the second moments, $\mathbf{rr}^\top$ (Table 1). In this simple

example, the nuisance variable of the spatial phase ensures that quadratic statistics contain relevant information about the orientation, just like complex cells in V1[4].

**Choice correlations predicted for optimal linear decoding.** To study how neural information is used or decoded, past studies have examined whether neurons that are sensitive to sensory inputs also reflect an animal's behavioral outputs or choices[39–47]. This choice-related activity is hard to interpret, because it may reflect decoding of the recorded neurons, or merely correlations between them and other neurons that are decoded instead[48]. However, testable predictions about the choice-related activity can reveal the brain's decoding efficiency for linear codes[27]. Next, we discuss these predictions, and then generalize them to nonlinear codes.

We define 'choice correlation' $C_{r_k}$ as the correlation coefficient between the response $r_k$ of neuron $k$ and the stimulus estimate (which we view as a continuous 'choice') $\hat{s}$, given a fixed stimulus $s$:

$$C_{r_k} = \text{Corr}(r_k, \hat{s}|s) \tag{2}$$

This choice correlation is a conceptually simpler and more convenient measure than the more conventional statistic, 'choice probability'[49], but it has almost identical properties (see the "Methods" subsection "Nonlinear choice correlations")[27,48].

Intuitively, if an animal is decoding its neural information efficiently, then those neurons encoding more information should be more correlated with the choice. Mathematically, one can show that choice correlations indeed have this property when decoding is optimal. There are several closely connected versions of this relationship that quantify information by estimator variance or Fisher information for continuous estimates, or by threshold or discriminability for binary estimates[27]—but the simplest is based on discriminability:

$$C_{r_k}^{\text{opt}} = \frac{d'_{r_k}}{d'} \tag{3}$$

where $d'$ and $d'_{r_k}$ are, respectively, the stimulus discriminability[50] based on the behavior or on neuron $k$'s response $r_k$ (see the "Methods" subsection "Nonlinear choice correlations"). This relationship holds for binary classification derived from a locally

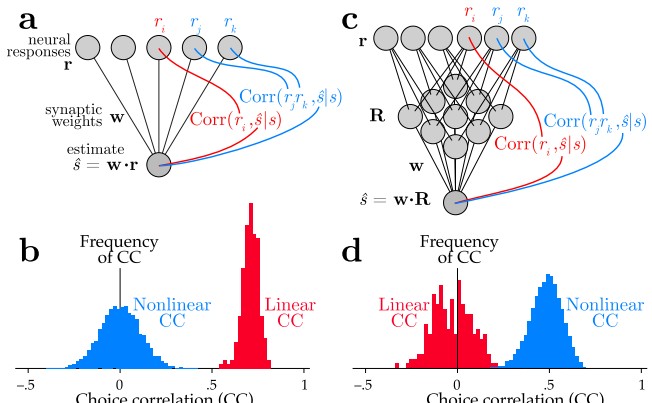

**Fig. 3 Linear and nonlinear choice correlations successfully distinguish network structure.** A linearly decoded population (**a**) produces nonzero linear choice correlations (**b**), while the nonlinear choice correlations are randomly distributed around zero. The situation is reversed for a nonlinear network (**c**), with insignificant linear choice correlations but strong nonlinear ones (**d**). Here the network implements a quadratic nonlinearity, so the relevant choice correlations are quadratic as well, $C_{jk} = \mathrm{Corr}(r_j r_k, \hat{s}|s)$.

optimal linear estimator

$$\hat{s} = \mathbf{w} \cdot \mathbf{r} + c \qquad (4)$$

for any stimulus-independent noise correlations, regardless of their structure.

Another way to test for optimal linear decoding would be to measure whether the animal's behavioral discriminability matches the discriminability for an ideal observer of the neural population response. Yet this approach is not feasible, as it requires one to measure simultaneous responses of many, or even all, relevant neurons, with enough trials to reliably estimate their joint information content. In contrast, the optimality test (Eq. (3)) requires measuring only non-simultaneous single neuron responses, which is vastly easier. Neural recordings in the vestibular system are consistent with near-optimal decoding according to this prediction[27].

**Nonlinear choice correlations for optimal decoding.** When nuisance variables wash out the mean tuning of neuronal responses, we may well find that a single neuron has both zero choice correlation and zero information about the stimulus. The optimality test would thus be inconclusive.

This situation is exactly the same one that gives rise to nonlinear codes. A natural generalization of Eq. (3) can reveal the quality of neural computation on nonlinear codes. We simply define a 'nonlinear choice correlation' between the stimulus estimate $\hat{s}$ and nonlinear functions of neural activity $\mathbf{R}(\mathbf{r})$:

$$C_{R_k} = \mathrm{Corr}(R_k(\mathbf{r}), \hat{s}|s) \qquad (5)$$

(see the "Methods" subsection "Nonlinear choice correlations"), where $R_k(\mathbf{r})$ is a nonlinear function of the neural responses. If the brain optimally decodes the information encoded in the nonlinear statistics of neural activity, according to the simple nonlinear extension to Eq. (4),

$$\hat{s} = \mathbf{w} \cdot \mathbf{R}(\mathbf{r}) + c \qquad (6)$$

then the nonlinear choice correlation satisfies the equation

$$C_{R_k(\mathbf{r})}^{\mathrm{opt}} = \frac{d'_{R_k(\mathbf{r})}}{d'} \qquad (7)$$

where $d'_{R_k(\mathbf{r})}$ is the stimulus discriminability provided by $R_k(\mathbf{r})$ (see the "Methods" subsection "Optimality test"). *This simple*

*Equation* (7) *is the most important in this paper, and it is the basis of most predictions and intuitions we present in subsequent sections.*

Equation (7) predicts that choice correlations of individual statistics will be stronger for more informative statistics, those with higher discriminability $d'_{R_k}$. This reflects either stronger stimulus tuning of the statistic and/or lower variability of that statistic. This effect does not depend on whether the variability is shared. The variability that is not decoded dilutes the choice correlation; variability that is decoded increases it. Finally, choice correlations for optimal decoding can never be negative: if a statistic is tuned to increase with the stimulus, its fluctuations should correlate with choices that increase as well.

As an example of this relationship, we return to the orientation task. Here the response covariance $\Sigma(s) = \mathrm{Cov}(\mathbf{r}|s)$ depends on the stimulus, but the mean $\mathbf{f} = \langle \mathbf{r}|s \rangle = \langle \mathbf{r} \rangle$ does not. In this model, optimally decoded neurons would have no linear correlation with behavioral choice. Instead, the choice should be driven by the product of the neural responses, $\mathbf{R}(\mathbf{r}) = \mathrm{vec}(\mathbf{r}\mathbf{r}^\top)$, where $\mathrm{vec}(\cdot)$ is a vectorization that flattens an array into a one-dimensional list of numbers. Such quadratic computation is what the energy model for complex cells is thought to accomplish for phase-invariant orientation coding[4]. Figure 3 shows linear and nonlinear choice correlations for pairs of neurons, defined as $C_{r_i r_j} = \mathrm{Corr}(r_i r_j, \hat{s}|s)$. When decoding is linear (a suboptimal strategy for this example), linear choice correlations are strong while nonlinear choice correlations are near zero (Fig. 3a, b). When the decoding is quadratic, here mediated by an intermediate layer that multiplies pairs of neural activity, the nonlinear choice correlations are strong while the linear ones are insignificant (Fig. 3c, d).

**Redundant codes.** It might seem unlikely that the brain uses optimal, or even near-optimal, nonlinear decoding. Even if it does, there are an enormous number of high-order statistics for neural responses, so the information content in any one statistic could be tiny compared to the total information in all of them. For example, with $N$ neurons there are on the order of $N^2$ quadratic statistics, $N^3$ cubic statistics, and so on. With so many statistics contributing information, the choice correlation for any single one would then be tiny according to the ratio in Eq. (7), and would be indistinguishable from zero with reasonable amounts of data. Past theoretical studies have described nonlinear (specifically, quadratic) codes with extensive information that grows proportionally with the number of neurons[2,28]. This would indeed imply immeasurably small choice correlations for large, optimally decoded populations.

A resolution to these concerns is information-limiting correlations[3]. The past studies that derive extensive nonlinear information treat large cortical populations in isolation from the smaller sensory population that would naturally provide its input[2,28]. Yet when a network inherits information from a much smaller input population, the expanded neural code becomes highly redundant: the brain cannot have more information than it receives[51]. Noise in the input is processed by the same pathway as the signal, and this generates noise correlations that can never be averaged away[3].

The previous work[3] characterized linear information-limiting correlations for fine discrimination tasks by decomposing the noise covariance into $\Sigma = \Sigma_0 + \epsilon \mathbf{f}' \mathbf{f}'^\top$, where $\epsilon$ is the variance of the information-limiting component and $\Sigma_0$ is noise that can be averaged away with many neurons.

For *nonlinear* population codes, it is not just the mean responses that encode the signal, $\mathbf{f}(s) = \langle \mathbf{r}|s \rangle$, but rather the nonlinear statistics $\mathbf{F}(s) = \langle \mathbf{R}(\mathbf{r})|s \rangle$. Likewise, the noise does not

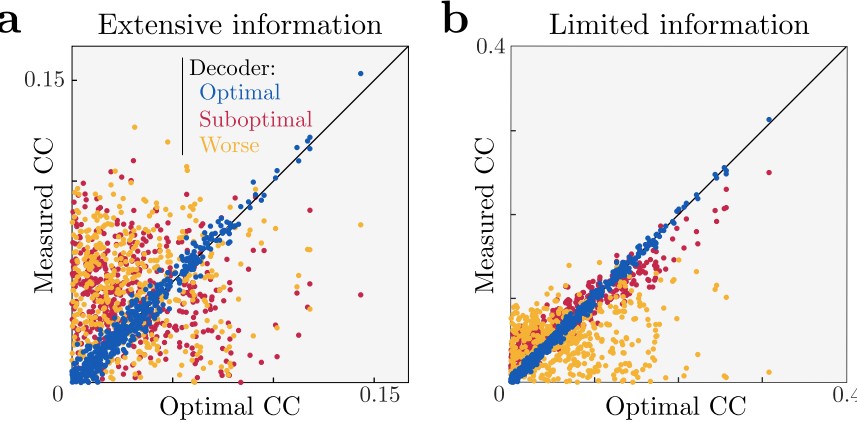

**Fig. 4 Information-limiting noise makes a network more robust to suboptimal decoding. a** A simulated optimal decoder produces measured choice correlations that match our optimal predictions (blue, on diagonal). In contrast, when a noise covariance $\Gamma_0$ permits the population to have extensive information, then a suboptimal decoder can exhibits a pattern of choice correlations that does not match the prediction of optimal decoding. Here we show two suboptimal decoders, one that is blind to higher-order correlations ($\mathbf{w} \propto \mathbf{F}'$, red), and another 'worse' decoder that has the same weights but with 40% random sign flips (green). As in Fig. 5, horizontal axis shows optimal choice correlations (Eq. (7)) and vertical axis shows measured choice correlations (Eq. (5)). **b** When information is limited, the same decoding weights may be less detrimental, and thus exhibit a similar pattern of choice correlations as an optimal decoder (red), or if they are sufficiently bad they may retain a suboptimal pattern of choice correlations (green).

comprise only second-order covariance of $\mathbf{r}$, $\mathrm{Cov}(\mathbf{r}|s)$, but rather the second-order covariance of the relevant nonlinear statistics, $\Gamma = \mathrm{Cov}(\mathbf{R}(\mathbf{r})|s)$ (see the "Results" subsection "Signal and noise"). Analogous to the linear case, these correlations can be locally decomposed as

$$\Gamma = \mathrm{Cov}(\mathbf{R}(\mathbf{r})|s) = \Gamma_0 + \epsilon\mathbf{F}'\mathbf{F}'^{\top} \qquad (8)$$

where $\epsilon$ is again the variance of the information-limiting component, and $\Gamma_0$ is any other covariance that can be averaged away in large populations, including internal noise and external nuisance variation. The information-limiting noise bounds the estimator variance $\sigma_{\hat{s}}^2$ to no smaller than $\epsilon$ even with optimal decoding. Likewise, the Fisher information cannot exceed the value of $1/\epsilon$, and the discriminability $d'$ cannot exceed $ds/\sqrt{\epsilon}$ for a stimulus change of $ds$[3]. Neither additional cortical neurons nor additional decoded statistics can improve performance beyond this bound.

**Consequences of optimal decoding on choice correlations.** The simple formula of Eq. (7) provides useful insights into the relationship between neural activity and choice when that activity is decoded optimally in a natural task. First, the choice correlations do not depend on the shape or magnitude of the internal noise or nuisance correlations, because these dimensions are deliberately avoided by optimal decoding, whose weights cancel those correlations (Eq. (18)). The only aspect of shared variability that matters for choice is the information-limiting component, i.e. that which is indistinguishable from a change in the task-relevant stimulus. This information-limiting variability increases choice correlations mostly by decreasing the overall behavioral discriminability $d'$; the shared variability is large only at the population level and typically only has a small contribution to any one statistic[3,27] and thus to its discriminability $d'_k$. With a smaller denominator and roughly unaffected numerator, the ratio in Eq. (7) rises with information-limiting correlations.

Likewise, the total number of the decoded statistics only affects the prediction of Eq. (7) insofar as it affects the available information. This is especially important when considering nonlinear statistics, as there are potentially so many of them. When optimally decoded, greater numbers of *independently* informative statistics will increase the total discriminability

exhibited by the behavioral choices, while the discriminability for each statistic remains unchanged. According to Eq. (7), choice correlations for optimal decoding are the ratio of these two, so as the behavioral behavioral $d'$ increases and the individual terms remain fixed, the choice correlations shrink. On the other hand, greater numbers of *redundant* statistics change neither the total information content nor the behavioral choice. These added statistics can have similar tuning and similar fluctuations as each other (which is what makes them redundant). Any single redundant statistic might or might not be decoded, but it is correlated with others that are. According to Eq. (7), the individual $d'_{R_k}$ are unchanged when adding more redundant statistics; the total information $d'$ is unchanged; and thus their ratio is fixed, consistent with the unchanged choice correlations.

When there are many fewer sensory inputs than cortical neurons, as seen in the brain, many distinct statistics $R_k(\mathbf{r})$ will carry redundant information. Under these conditions, many choice correlations $C_{R_k}$ can be quite large even for optimal nonlinear decoding: the discriminabilities $d'_{R_k}$ of redundant statistics can be comparable to the discriminability $d'$ of the whole population, producing ratios $d'_{R_k}/d'$ that are a significant fraction of 1 (Fig. 4). Supplementary Information S.7.1 illustrates this effect for a redundant nonlinear population code: the brain need not decode all functions of all neurons to extract essentially all of the information (Fig. S6A), and neuroscientists need not compute choice correlations for all possible statistics to establish decoding efficiency (Fig. S6B).

**Which nonlinear statistics?** If the brain's decoder optimally uses all available information, choice correlations will obey the prediction of Eq. (7) even if the specific nonlinear statistics extracted by the brain's decoder differ from those selected for evaluating choice correlations (see the "Methods" subsection "Nonlinear choice correlation to analyze an unknown nonlinearity"). The prediction is valid as long as the brain's nonlinearity can be expressed as a linear combination of the tested nonlinearities (see the "Methods" subsection "Nonlinear choice correlation to analyze an unknown nonlinearity"). Since the brain needs complicated nonlinearities for complicated tasks, it may be difficult to find a suitable basis set for truly natural conditions; feature spaces from deep networks trained on comparable tasks might provide a

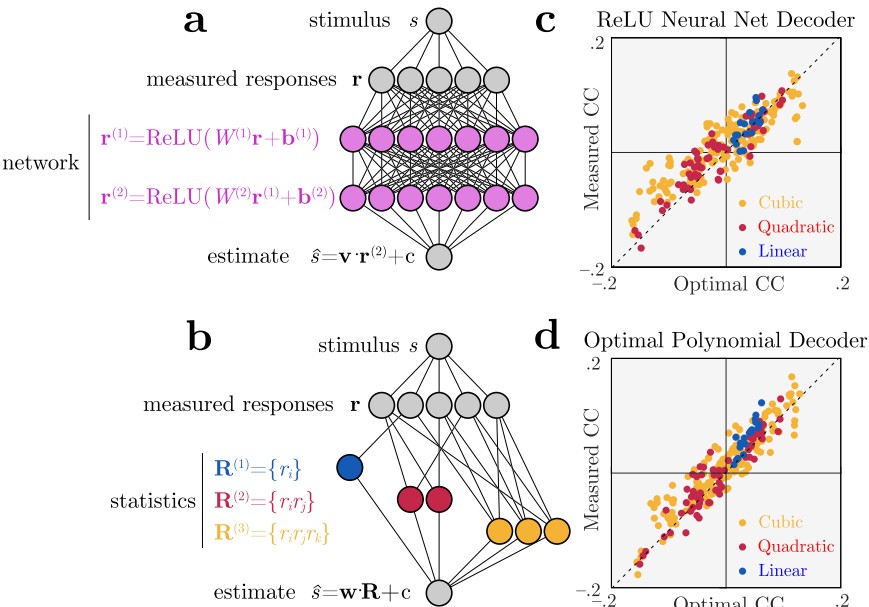

**Fig. 5 Identifying optimal nonlinear decoding by a generic neural network using nonlinear choice correlations.** Neural responses **r** are constructed to encode stimulus information in polynomial sufficient statistics up to cubic order (see the "Methods" section Eq. (13)). These responses are decoded by an artificial nonlinear neural network or polynomial nonlinearities, and we evaluate the quality of the decoding using polynomial nonlinearities for both cases. **a** Architecture of a network that uses ReLU nonlinearities trained to extract the relevant information. **b** Architecture of a second network that instead uses polynomial nonlinearities to extract the relevant information. **c**, **d** Choice correlations based on polynomial statistics show that both networks' computations are consistent with optimal nonlinear decoding (see the "Methods" subsection "Nonlinear choice correlation to analyze an unknown nonlinearity"), even though the simulated networks used different implementations to extract the stimulus information. Horizontal axis shows optimal choice correlations (Eq. (7)); vertical axis shows measured choice correlations (Eq. (5)).

useful basis[38,52]. For the modest, controlled-complexity tasks used in most neuroscience experiments[53–56], polynomials or other simple bases may be sufficient, even when individual neurons do not use polynomial operations.

Indeed, Fig. 5 shows a situation where information is encoded by linear, quadratic and cubic sufficient statistics of neural responses, but a simulated brain decodes them near-optimally using a generic neural network rather than a set of nonlinearities matched to those sufficient statistics. Despite this mismatch we can successfully identify that the brain is near-optimal by applying Eq. (7), even without knowing details of the simulated brain's true nonlinear transformations.

**Decoding efficiency revealed by choice correlations.** Even if decoding is not strictly optimal, Eq. (7) can be approximately satisfied due to information-limiting correlations. Decoders that seem substantially suboptimal because they fail to avoid the largest noise components in $\Gamma_0$ can be nonetheless dominated by the bound from information-limiting correlations. This will occur whenever the variability from suboptimally decoding the noise $\Gamma_0$ is smaller than the information-limiting variance $\epsilon$. Just as we can decompose the nonlinear noise correlations into information-limiting and other parts, we can decompose nonlinear choice correlations into corresponding parts as well, with the result that

$$C_R^{\text{sub}} \approx \alpha C_R^{\text{opt}} + \chi_R \qquad (9)$$

where $\chi_R$ depends on the particular type of suboptimal decoding (Supporting Information S.3.2). The slope $\alpha$ between choice correlations and those predicted from optimality is given by the fraction of estimator variance explained by information-limiting noise, $\alpha = \epsilon/\sigma_{\hat{s}}^2$. This slope $\alpha$ therefore provides an estimate of the efficiency of the brain's decoding.

Figure 4 shows an example of one decoder that would be suboptimal without redundancy, but is nonetheless close to

optimal when information limits are imposed. This rescue of optimality does not happen for all decoders, however. The figure also shows another decoder that is so suboptimal that it throws away most of the available information even when there is substantial redundancy. The patterns of choice correlations reflect this.

In realistically redundant models with more cortical neurons than sensory inputs, many decoders could be near-optimal, as we recently discovered in experimental data for a linear population code[27]. However, even in redundant codes there may be substantial inefficiencies and information loss[57], especially for unnatural tasks[58], so it is scientifically interesting to discover near-optimal nonlinear computation even in a redundant code.

**Coarse versus fine, estimation versus classification.** Our conceptual framework and predictions are most simply expressed for fine estimation tasks, where here we define 'fine' as a stimulus range over which the noise statistics do not vary with the stimulus. Some minor details of our predictions change when moving to binary classification instead of continuous estimation: this introduces a correction factor that depends on the response distribution (Section S.6.4).

More details change when moving to 'coarse' tasks, which we define as when noise statistics do change significantly with the stimulus. As for fine discrimination, we again find that when decoding is optimal, random fluctuations in choices are correlated with neural responses to the same degree that those responses can discriminate between stimuli. However, this relationship is slightly more complicated for coarse discrimination. For this reason we introduce a slightly more complicated measure of choice correlation that we call Normalized Average Conditional Choice Correlation (NACCC, Eq. (17)), which removes the stimulus-induced covariation between neuron and choice, and isolates only the remaining shared fluctuations that reflect the

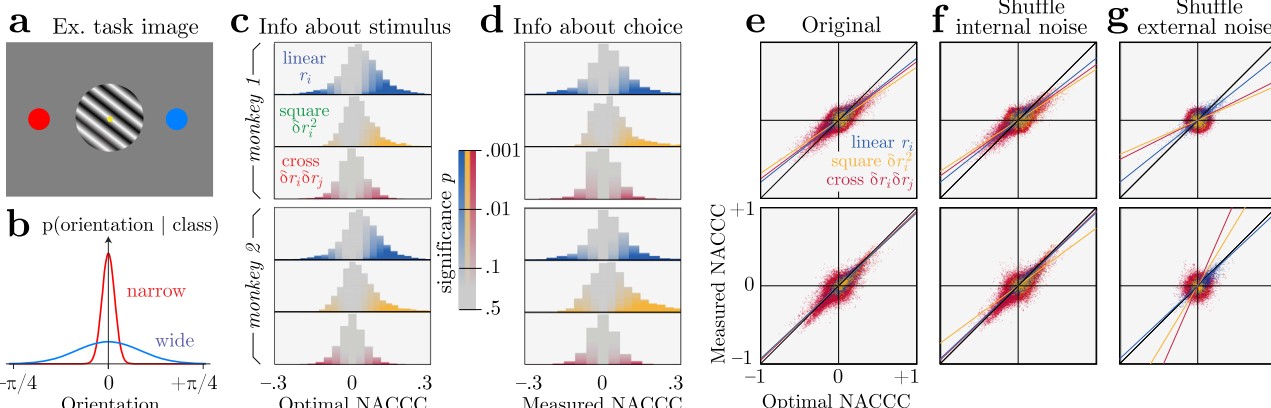

**Fig. 6 Nonlinear information and choice correlations in a variance discrimination task, for neural data from two monkeys. a** Example oriented grating and saccade targets. **b** The orientations of the gratings were drawn from a narrow or wide distribution, and the monkey had to guess which by saccading to the appropriate target. **c** Neurons contain linear and nonlinear information about the task variable. This is revealed by the Normalized Average Conditional Choice Correlations (NACCC, Eq. (17)) predicted for *optimal* decoding, which are proportional to the measured signal-to-noise ratios (Eq. (7)) for each neural response pattern (blue $r_i$, green $\delta r_i^2$, red $\delta r_i \delta r_j$). Color saturation indicates statistical significance (see the "Methods" subsection "Application to neural data"). **d** These neurons also contain significant information about the animal's choice, as computed by the *measured* NACCC. **e** The measured and optimal NACCCs are highly correlated, with a proportionality near 1 (lines). The coefficient of determination, R-squared is 0.50, 0.33, 0.12 for linear, square and cross terms for monkey 1; 0.61, 0.64, 0.40 for monkey 2. Each point represents one response pattern (e.g. $\delta r_i \delta r_j$) in one session. Top and bottom panels are data from two different monkeys. These two plotted quantities are strongly correlated (0.76, 0.65, 0.53 for linear, square and cross terms for monkey 1; 0.80, 0.83, 0.72 for monkey 2. **f** Shuffling internal noise correlations while preserving nuisance correlations maintains the relationship between prediction and nonlinear choice correlations, implying that internal noise is not responsible for the correlations. **g** Shuffling nuisance correlations across trials (see the "Methods" subsection "Application to neural data") nearly eliminates the relationship between measured and predicted nonlinear choice correlations (0.76, 0.05, 0.04 for monkey 1; 0.80, 0.10, 0.11 for monkey 2), implying that nuisance variation creates the nonlinear code.

brain's processing. However, the end result is the same: choice correlations for optimal decoding are equal to the ratio of discriminabilities (Eq. (7); Supplemental Information S.5). As for fine estimation, there is a correction factor of order 1 for binary choices instead of continuous estimation (see the "Methods" subsection "Optimality test", Supplemental Information S.6.4, Eq. (185)).

**Evidence for optimal nonlinear computation in macaque brains**. We applied our optimality test to data recorded with Utah arrays from primate visual cortex (V1) during a nonlinear decoding task. Monkeys performed a Two-Alternative Forced Choice task (2AFC) in which they categorized an oriented drifting grating based on whether it came from a wide or narrow distribution of orientations[59] (Fig. 6a, b). The categorical target variable $s$ is therefore the variance of the orientation distribution. This coarse binary discrimination task is a simplified version of a task that might arise in nature when identifying a surface texture or material[60]; the orientation of the material would be a nuisance variable independent of the material type. Here the observable variable, the orientation, is the product of the target variable and a nuisance variable $v$. An additional nuisance variable was the stimulus contrast varying independently of the stimulus variance, although here we only analyze the highest contrast.

Below we analyze whether the trial-by-trial nonlinear statistics of V1 multi-unit neural responses to these stimuli provide information about the task-relevant category and the behavioral choice, and whether these two informations are correlated as predicted by our optimal decoding theory. Since the marginal response statistics depend substantially on the stimulus category, at least in part because the corresponding nuisance distributions differ, this is a coarse binary discrimination task. For this reason we test suitable optimal decoding predictions about nonlinear choice correlations in coarse tasks, using the NACCC measure that we outlined in the section "Coarse versus fine, estimation

versus classification" and derived in the "Methods" subsection "Application to neural data" and Supplemental Information S.6.

V1 responses contain information about orientation[61]. Here we found that V1 responses also contain some linear information about the orientation *variance* (Fig. 6c, blue; $d'$ calculated by Eq. (20)). This implies that within their receptive field they have already performed some nonlinear transformations of the input that are useful for estimating the orientation variance. However, we expect that nonlinear computations downstream can extract still more information. Note that for a fixed contrast, an optimal computation based on the stimulus is simply to threshold the squared deviation from the mean orientation. Because neural responses in this brain area can be linearly decoded to compute orientation, a good downstream decoder for the orientation variance would naturally be quadratic in those responses.

Indeed, we found information in the quadratic statistics of neural responses, $\delta r_i^2$ and $\delta r_i \delta r_j$ (Fig. 6c, red and green), verifying that downstream nonlinear computations could extract additional information from the neural responses. To isolate the nonlinear information we eliminated the linear stimulus dependence of the response, computing neural nonlinear statistics according to $\delta r_i = r_i - \langle r_i | \hat{s}_1 \rangle$, where $\hat{s}_1 = \mathbf{w}_{\text{opt}} \cdot \mathbf{r} + c$ is the optimal estimate decoded only from a linear combination of available neural responses.

These quadratic statistics also contained substantial nonlinear information about the behavioral choice (Fig. 6d). In general, there is no guarantee that the particular nonlinear statistics that are informative about the stimulus are also informative about the choice. However, our theory of optimal decoding predicts specifically that these quantities should be directly proportional to each other. Indeed, in two monkeys, we found that nonlinear choice correlations were highly correlated with nonlinear stimulus information (Fig. 6e). Remarkably, when we compare the measured nonlinear choice correlations to the ratio of discriminabilities after adjusting for the binary data (see the

"Methods" subsection "Application to neural data"), the slopes of this relationship for the two animals were near the value of 1 that Eq. (7) predicts for optimal decoding (Fig. 6e).

Monkey 2 performed slightly worse than an ideal observer, with a probability correct of 0.76, compared to the ideal of 0.82 (see the "Methods" subsection "Application to neural data")—even while its decoding was near-optimal, with an efficiency according to Eq. (9) of $0.96 \pm 0.04$ (mean ± 95% confidence intervals). Even at the level of individual sessions, this is consistent with optimal decoding, with efficiencies not significantly different from 1 ($p = 0.26$, one-tailed $t$-test). This suggests that information is lost in the encoding stage somewhere between the stimulus and the recorded neurons, and not downstream of those neurons. Monkey 1 had similar overall performance (probability correct of 0.74) but worse decoding efficiency ($0.75 \pm 0.08$). Across sessions with reliable slopes (positive coefficients of determination, 77/119 sessions), the efficiencies were significantly different from 1 ($p < 10^{-6}$, one-tailed $t$-test). This suggests the second monkey's task performance has limitations arising downstream of the recorded neurons.

**Controls to find the origins of choice correlations**. To evaluate whether internal noise correlations contribute nonlinear information or choice correlations, we created a shuffled data set that removed internal noise correlations while preserving external nuisance correlations. That is, for each neuron we independently selected responses to high-contrast trials with matched target stimulus (variance), nuisance (orientations within ±1.5), and choice, and repeated our analysis on these shuffled data (Fig. 6f). The observed relationship between predicted and observed choice correlations was the same as in the original test, indicating that nuisance variations were sufficient to drive the nonlinear information and decoding.

We then shuffled the external nuisance correlations by randomly selecting responses to trials with matched target stimulus and choice, but now using *unmatched* nuisance variables, and again repeated the analysis (Fig. 6g). In other words, we picked responses from different trials that came from the same signal category (wide or narrow) and elicited the same choice but had different orientations, and we picked these trials (and thus their stimulus orientations) *independently* for neurons $i$ and $j$. The strong statistical relationship observed between predicted and measured nonlinear choice correlations vanished with this shuffling, indicating that the nuisance variation was necessary for the nonlinear information and nonlinear decoding.

These shuffle controls removed noise correlations and nuisance correlations, respectively. Combining the conclusions from these controls, we find no evidence that the brain optimally decodes any stimulus-dependent internal noise correlations in this task.

We looked directly for stimulus-dependent internal noise correlations by conditioning on both the signal and the nuisance variable (which here is simply the single number, orientation) and measuring orientation-dependent response covariances. The resultant nonlinear tuning was quite weak compared with the trial-to-trial variability in those nonlinear statistics, and available nonlinear information arose largely in changing variances rather than covariances; likely arising from Poisson statistics and tuning of the mean firing (Supplementary Fig. S5A, B). Internal noise fluctuations in those directions were not significantly correlated with choice (Supplementary Fig. S5C, $p = 0.088, 0.830, 0.969$ for linear, square, and cross terms for monkey 1; $p = 0.073, 0.094, 0.573$ for linear, square, and cross terms for monkey 2 using a two-sample Kolmogorov–Smirnov test).

Recent analyses of these same data found that internal noise did in fact influence the monkeys' behavioral choices[62], but this effect was subtle and only apparent when examining the entire neural population simultaneously with a complex trained nonlinearity. In our analysis this effect is buried in the noise, so our method is not sensitive enough to tell if these large-scale patterns induced by internal noise are used optimally or suboptimally. Additionally, in this work we analyzed a subset of trials with the highest contrasts and it is possible that at lower contrasts internal noise has a greater influence. However, we can detect that the brain contains information that is encoded nonlinearly due to external nuisance variation, and that this information is indeed decoded near-optimally by the brain.

## Discussion

This study introduced a theory of nonlinear population codes, grounded in the natural computational task of separating relevant and irrelevant variables. The theory considers both encoding and decoding—how stimuli drive neurons, and how neurons drive behavioral choices. It shows how correlated fluctuations between neural activity and behavioral choices could reveal the efficiency of the brain's decoding. Unlike previous theories of nonlinear population codes[2,28], ours remains consistent with biological constraints due to the large cortical expansion of sensory representations by incorporating redundancy through a nonlinear generalization of information-limiting correlations[3]. Also unlike past work which largely concentrates on *encoding* efficiency, we provide mathematical methods to quantify the brain's nonlinear *decoding* efficiency. When we applied this method to the neural responses of monkeys performing a discrimination task in which neural statistics were dominated by nuisance variation, we found quantitative results consistent with efficient nonlinear decoding of V1 activity.

The best condition to apply our optimality test is in a task of modest complexity but still possessing fundamentally nonlinear structure. Some interesting examples where our test could have practical relevance include motion detection using photoreceptors[63], visual search with distractors (XOR-type tasks)[30,64], sound localization in early auditory processing before the inferior colliculus[65], or context switching in higher-level cortex[55].

Optimal nonlinearities extract the sufficient statistics about the relevant stimulus. These statistics depend not only on the task but also on the nuisance variables. In complex tasks, like recognizing objects from images, nuisance variables push most of the relevant information into higher-order statistics which require more complex nonlinearities to extract. In such high-dimensional cases, our proposed test is unlikely to be useful. This is because our method expresses stimulus estimates as sums of nonlinear functions, and while that is universal in principle[66], that is not a compact way to express the complex nonlinearities of deep networks. Relatedly, it may be difficult to see statistically significant information or choice correlations for nonlinear statistics that provide many important but small contributions to the behavioral output. Since many stimulus-dependent response correlations are induced by external nuisance variation, not internal noise, we might not find informative stimulus-dependent noise correlations upon repeated presentations of a fixed stimulus. Indeed, our analysis found no evidence of internal noise generating nonlinear choice correlations (Fig. 6). Those correlations may only be informative about a stimulus in the presence of natural nuisance variation. For example, if a picture of a face is shown repeatedly without changing its pose, then small expression changes can readily be identified by linear operations; if the pose varies then the stimulus is only reflected in higher-order correlations[9].

In contrast, we *should* see some nonlinear choice correlations even when nuisance variables are fixed. This is because neural circuitry must combine responses nonlinearly to eliminate natural

nuisance variation, and any internal noise passing through those same channels will thereby influence the choice. Although they may be smaller and more difficult to detect than the fluctuations caused by the nuisance variation, this influence will manifest as nonlinear choice correlations. In other words, nonlinear noise correlations need not predict a fixed stimulus, but they may predict the choice (Supplementary Information S.4).

Our approach is currently limited to spatial feedforward processing, which unquestionably oversimplifies cortical processing. The approach can be generalized to recurrent networks by considering spatiotemporal statistics[67]. Feedback could also cause suboptimal networks to exhibit choice correlations that seem to resemble the optimal prediction. If the feedback is noisy and projects into the same direction that encodes the stimulus, such as from a dynamic bias[68–70], then this could appear as information-limiting correlations, enhancing the match with Eq. (7). This situation could be disambiguated by measuring the internal noise source providing the feedback, though of course this would require more simultaneous measurements.

In principle our method can also be applied to temporal neural response properties like spike timing or time series. For optimal processing, spike timing that is tuned to task-relevant stimuli should also be correlated with resultant choices, even if the timing is converted to rate codes by downstream processing. On the other hand, it is more difficult to track behavioral consequences of spatiotemporal correlations that evolve through a recurrent network[67] with dynamic outputs, as in motor control applications. It should be fruitful to develop this theory further for more complex tasks involving time sequences of actions.

Our method to understand nonlinear neural decoding requires neural recordings in a behaving animal. The task must be hard enough that it makes some errors, so that there are behavioral fluctuations to explain. Finally, there should be a modest number of nonlinearly entangled nuisance variables. Unfortunately, many neuroscience experiments are designed without explicit use of nuisance variables. Although this simplifies the analysis, this simplification comes at a great cost, which is that the neural circuits are being engaged far from their natural operating point, and far from their purpose: there is little hope of understanding neural computation without challenging the neural systems with nonlinear tasks for which they are required. In this context, it is especially noteworthy that a mismatch between choice correlations and the optimal pattern might not indicate that the brain is suboptimal, but instead that the nuisance variation in the experimental task may not match the natural tasks the brain has learned. For this reason it is important for neuroscience to use natural tasks, or at least naturalistic ones, when aiming to understand computational function[71–73].

## Methods

**Orientation estimation with varying spatial phase**. Figure 1 illustrates how nuisance variation can eliminate a neuron's mean tuning to relevant stimulus variables, relegating the neural tuning to higher-order statistics like covariances. In this example, the subject estimates the orientation of a Gabor image, $G(\mathbf{x}|s, v)$, where $\mathbf{x}$ is spatial position in the image, and $s$ and $v$ are the orientation and spatial phase (nuisance) of the image, respectively (Supplementary Material S.1.1). The model visual neurons are linear Gabor filters like idealized simple cells in primary visual cortex, corrupted by additive white Gaussian noise. Their responses are thus distributed as $\mathbf{r} \sim P(\mathbf{r}|s, v) = N(\mathbf{r}|\mathbf{f}(s, v), \epsilon I)$, where $\epsilon$ is the noise variance and the mean $\mathbf{f}(s, v) = \langle \mathbf{r}|s, v \rangle = \sum_{\mathbf{r}} \mathbf{r}\, p(s|\mathbf{r}, v)$ is determined by the overlap between the image and the receptive field.

When the spatial phase $v$ is known, the mean neural response contains all the information about orientation $s$. The brain can decode responses linearly to estimate orientation near a reference $s_0$.

When the spatial phase varies, however, each mean response to a fixed orientation will be combined across different phases:
$\mathbf{f}(s) = \langle \mathbf{r}|s \rangle = \sum_{\mathbf{r}} \mathbf{r}\, p(\mathbf{r}|s) = \int d v \sum_{\mathbf{r}} \mathbf{r}\, p(\mathbf{r}|s, v) p(v)$. Since each spatial phase can be paired with another phase $\pi$ radians away that inverts the linear response, the

phase-averaged mean is $\mathbf{f}(s) = 0$. Thus the brain cannot estimate orientation by decoding these neurons linearly; nonlinear computation is necessary.

The covariance provides one such tuned statistic. We define $\mathrm{Cov}_{ij}(\mathbf{r}|s, v)$ as the neural covariance for a fixed input image (noise correlations), and $\mathrm{Cov}_{ij}(\mathbf{r}|s)$ as the neural covariance when the nuisance varies (nuisance correlations). According to the law of total covariance,

$$\mathrm{Cov}_{ij}(\mathbf{r}|s) = \int d v\, (\mathrm{Cov}_{ij}(\mathbf{r}|s, v) + \delta f_i(s, v)\delta f_j(s, v))p(v) \tag{10}$$

where $\delta f_i(s, v) = f_i(s, v) - \langle f_i(s, v) \rangle_v$. Supplementary Information S.1.1 shows in detail how $\mathrm{Cov}_{ij}(\mathbf{r}|s)$ is tuned to $s$.

**Exponential family distribution and sufficient statistics**. It is illuminating to assume the response distribution conditioned on the relevant stimulus (but not on nuisance variables) is approximately a member of the exponential family with nonlinear sufficient statistics,

$$p(\mathbf{r}|s) = b(\mathbf{r}) \exp(\mathbf{H}(s) \cdot \mathbf{R}(\mathbf{r}) - A(s)) \tag{11}$$

where $\mathbf{R}(\mathbf{r})$ is a vector of sufficient statistics for the natural parameter $\mathbf{H}(s)$, $b(\mathbf{r})$ is the base measure, and $A(s)$ is the log-partition function. In this case, a finite number of sufficient statistics contains all of the information about the stimulus in the population response, and all other tuned statistics may be derived from them.

Estimation and inference are closely connected in the exponential family. In Supplementary Material S.1.2.2, we show that the optimal local estimation can be achieved by linearly decoding the nonlinear sufficient statistics, $\hat{s} = \mathbf{w}^\top \mathbf{R}(\mathbf{r}) + c$. The decoding weights minimize the variance of an unbiased decoder,

$$\mathbf{w}_{\mathrm{opt}} \propto \mathbf{H}'(s) \propto \Gamma^{-1}\mathbf{F}' \tag{12}$$

where $\mathbf{F}' = \partial\langle \mathbf{R}(\mathbf{r})|s \rangle / \partial s$ is the sensitivity of the statistics to changing inputs, and $\Gamma = \mathrm{Cov}(\mathbf{R}|s)$ is the stimulus-conditioned response covariance—which generally includes nuisance correlations (see the section "Signal and noise").

**Quadratic encoding**. In a quadratic coding model, the distribution of neural responses is described by the exponential family with up to quadratic sufficient statistics, $\mathbf{R}(\mathbf{r}) = \{r_i, r_i r_j\}$ for $i, j \in \{1, ..., N\}$. A familiar example is the Gaussian distribution with stimulus-dependent covariance $\Sigma(s)$. In order to demonstrate the coding properties of a purely nonlinear neural code, here we assume that the mean tuning curve $f(s)$ is constant, while the stimulus-conditional covariances $\Sigma_{ij}(s)$ depend smoothly on the stimulus. We can quantify the information content of the neural population using Eq. (61).

**Cubic encoding**. In our cubic coding model, the distribution of neural responses is described by the exponential family with up to cubic sufficient statistics, $\mathbf{R}(\mathbf{r}) = \{r_i, r_i r_j, r_i r_j r_k\}$ for $i, j, k \in \{1, ..., N\}$.

We approximate a three-neuron cubic code first using purely cubic components, and we then apply a stimulus-dependent affine transformation to include linear and quadratic statistics. The pure cubic code is used for a vector $\mathbf{z}$ with sufficient statistics $z_i z_j z_k$ (and a base measure $e^{-\|\mathbf{z}\|^4}$ to ensure the distribution is bounded and normalizable).

$$p(\mathbf{z}|s) = \frac{1}{Z}\exp\left(-\|\mathbf{z}\|^4 + \gamma s\, z_i z_j z_k\right) \tag{13}$$

We approximate this distribution by a mixture of four Gaussians. The mixture is chosen to reproduce the tetrahedral symmetry of the cubic distribution (Supplementary Fig. S1), which allows the cubic statistics of responses to be stimulus dependent, leaving stimulus-independent quadratic and linear statistics.

To generate larger multivariate cubic codes for Supplementary Fig. S1, for simplicity we assume the pure cubic terms only couple disjoint triplets of variables, and sample independently from an approximately cubic distribution for each triplet. To convert this purely cubic distribution to a distribution with linear and quadratic information, we shift and scale these cubic samples $\mathbf{z}$ in a manner dependent on $s$:

$$\mathbf{r} = \mathbf{f}(s) + \Sigma^{1/2}(s)\mathbf{z} \tag{14}$$

where $\mathbf{f}(s)$ and $\Sigma(s)$ describes the desired signal-dependent mean and covariance (see Supplementary Material S.1.4).

**Nonlinear choice correlations**. For fine discrimination tasks, the nonlinear choice correlation between the stimulus estimate $\hat{s} = \mathbf{w}^\top \mathbf{R} + c$ and one nonlinear function $R_k$ (the $k$th element of the vector $\mathbf{R}$) of recorded neural activity $\mathbf{r}$ is

$$C_{R_k} = \mathrm{Corr}(R_k(\mathbf{r}), \hat{s}|s) = \frac{(\Gamma\mathbf{w})_k}{\sqrt{\Gamma_{kk}\mathbf{w}^\top \Gamma\mathbf{w}}} \tag{15}$$

where $\mathbf{w}^\top \Gamma\mathbf{w} = \sigma_{\hat{s}}^2$ is the estimator variance.

When the relevant response statistics change appreciably over the stimulus range used in the task, such as for the coarse variance discrimination task in the section "Evidence for optimal nonlinear computation in macaque brains"), the relevant quantities change slightly. The optimal linear decoder of nonlinear

statistics, $\hat{s} = \mathbf{w} \cdot \mathbf{R} + c$, has weights obtained through linear regression:

$$\mathbf{w} \propto \bar{\Gamma}^{-1} \Delta \mathbf{F} \qquad (16)$$

where $\bar{\Gamma} = \langle \text{Cov}(\mathbf{R}|s) \rangle_s$ is the average conditional covariance between $\mathbf{R}$ given the stimulus $s$. The differences from Eq. (12) are $\Gamma \to \bar{\Gamma}$ and $\mathbf{F}' = d\mathbf{F}/ds \to \Delta\mathbf{F}/\Delta s$.

These differences are reflected in a slightly modified measure of correlation that we call normalized average conditional choice correlations (NACCC),

$$B_{R_k} = \frac{\langle \text{Cov}(R_k, \hat{s}|s) \rangle_s}{\sqrt{\langle \text{Var}(R_k|s) \rangle_s \langle \text{Var}(\hat{s}|s) \rangle_s}} = \frac{(\bar{\Gamma}\mathbf{w})_k}{\sqrt{\bar{\Gamma}_{kk}\mathbf{w}^\top \bar{\Gamma}\mathbf{w}}} \qquad (17)$$

$B_{R_k}$ is actually a correlation coefficient based on the average conditional covariance $\bar{\Gamma}$, and is bounded in absolute value by 1. As the stimulus range in a coarse task decreases, and the noise distribution $p(\mathbf{R}|s)$ becomes independent of the stimulus, then Eq. (17) converges toward Eq (15).

The choice correlation for binary choices differs slightly from that for continuous estimation, for both fine and coarse discrimination tasks, by a factor $\zeta$ that is typically of order 1 (Supplementary Materials S.6.1).

**Optimality test**. Substituting the optimal weights (Eq. (12)) into Eq. (15), the optimal nonlinear choice correlation becomes

$$C_{R_k(\mathbf{r})}^{\text{opt}} = \frac{(\Gamma \Gamma^{-1} \mathbf{F}')_k}{\sqrt{\Gamma_{kk} \mathbf{F}'^\top \Gamma^{-1} \mathbf{F}'}} = \frac{F_k'}{\sqrt{\Gamma_{kk}}} \sigma_{\hat{s}} = \frac{d'_{R_k(\mathbf{r})}}{d'} \qquad (18)$$

where $d'_{R_k(\mathbf{r})} = F_k' \Delta s / \sqrt{\Gamma_{kk}}$ is the fine discriminability provided by $R_k(\mathbf{r})$ for a stimulus difference of $\Delta s$. The same argument holds for coarse discrimination, where $\bar{\Gamma}$ in Eq. (17) is canceled by $\bar{\Gamma}^{-1}$ in the optimal weights (Eq. (16)), yielding $B_{R_k(\mathbf{r})}^{\text{opt}} = d'_{R_k} / d'$.

For fine-scale discrimination, optimal choice correlations can be written in many equivalent ways that reflect the simple relationships between four quantities often used to represent information: discriminability $d$-prime is proportional to the square root of the Fisher information $d' = \Delta s \sqrt{J}$ [74]; estimator variance is bounded by the inverse of the Fisher information, $\sigma_{\hat{s}}^2 \geq 1/J$; discrimination threshold is proportional to the estimator standard deviation, $\theta = \sqrt{\sigma_{\hat{s}}^2}$ with proportionality given by the threshold condition.

In different experiments (binary discrimination, continuous estimation), it can be most natural to express this optimal decoding prediction as ratios of different measured quantities:

$$C_{R_k}^{\text{opt}} = \frac{d'_{R_k}}{d'} = \frac{\theta}{\theta_{R_k}} = \sqrt{\frac{\sigma_{\hat{s}}^2}{\sigma_{\hat{s},R_k}^2}} = \sqrt{\frac{J_{R_k}}{J}} \qquad (19)$$

These quantities reflect information between the stimulus and the neural or behavioral responses. Supplemental material S.5 shows how this can be computed easily for general binary discrimination using the total correlation between the responses and the stimuli, $D_{R_k} = \text{Corr}(R_k, s)$, or a continuously varying behavioral choice $\hat{s}$ and the stimuli, $D_{\hat{s}} = \text{Corr}(\hat{s}, s)$:

$$d' = \frac{2}{\sqrt{D^{-2} - 1}} \approx 2D \qquad (20)$$

and likewise for $d'_{R_k}$. When the behavioral choice is binary rather than continuous, the correlations are modified by a factor $\delta$ near 1 (Supplemental Information S.6.3, Eq. (182)). For our experimental conditions, $\delta \approx 1.2 \pm 0.2$.

**Nonlinear choice correlation to analyze an unknown nonlinearity**. In Fig. 5, we generated neural responses given sufficient statistics that are polynomials up to third order, $\mathbf{R}(\mathbf{r}) = \{r_i, r_i r_j, r_i r_j r_k\}$ (see the "Methods" subsection "Cubic encoding"). Our model brain decodes the stimulus using a cascade of linear–nonlinear transformations, with Rectified Linear Units (ReLU$(x) = \max(0, x)$) for the nonlinear activation functions. We used a fully connected ReLU network with two hidden layers and 30 units per hidden layer. We trained the network weights and biases with backpropagation to estimate stimuli near a reference $s_0$ based on 20,000 training pairs $(\mathbf{r}, s)$ generated by the cubic encoding model. This trained neural network extracted 91% of the information available to an optimal decoder.

**Information-limiting correlations**. Only specific correlated fluctuations limit the information content of large neural populations[3]. These fluctuations can ultimately be referred back to the stimulus as $\mathbf{r} \sim p(\mathbf{r}|s + ds)$, where $ds$ is zero mean noise, whose variance $1/J_\infty$ determines the asymptotic variance of any stimulus estimator. These information-limiting correlations for nonlinear computation can be characterized by the covariance of the sufficient statistics, $\Gamma = \text{Cov}(\mathbf{R}|s)$ conditioned on $s$; the information-limiting component arises specifically from the signal covariance $\text{Cov}(\mathbf{F}(s)|s)$. Since the signal for local estimation of stimuli near a reference $s_0$ is $\mathbf{F}'(s) = \frac{d}{ds} \langle \mathbf{R}(\mathbf{r})|s \rangle$, the information-limiting component of the covariance is

proportional to $\mathbf{F}'\mathbf{F}'^\top$:

$$\Gamma = \Gamma_0 + \frac{1}{J_\infty} \mathbf{F}(s)' \mathbf{F}(s)'^\top \qquad (21)$$

Here $\Gamma_0$ is any covariance of $\mathbf{R}$ that does *not* limit information in large populations. Substituting this expression into (Eq. (61)) for the nonlinear Fisher Information, we obtain

$$J = \mathbf{F}'\Gamma^{-1}\mathbf{F}' = \frac{1}{1/J_\infty + 1/J_0} \qquad (22)$$

where $J_0 = \mathbf{F}'\Gamma_0^{-1}\mathbf{F}'$ is the nonlinear Fisher Information allowed by $\Gamma_0$. When the population size grows, the extensive information term $J_0$ grows proportionally, so the output information will asymptote to $J_\infty$.

**Application to neural data**. All behavioral and electrophysiological data were obtained from two healthy, male rhesus macaque (*Macaca mulatta*) monkeys (L and T) aged 10 and 7 years and weighting 9.5 and 15.1 kg, respectively. All experimental procedures complied with guidelines of the NIH and were approved by the Baylor College of Medicine Institutional Animal Care and Use Committee (permit number: AN-4367). Animals were housed individually in a room located adjacent to the training facility on a 12 h light/dark cycle, along with around 10 other monkeys permitting rich visual, olfactory, and auditory social interactions. Regular veterinary care and monitoring, balanced nutrition and environmental enrichment were provided by the Center for Comparative Medicine of Baylor College of Medicine. Surgical procedures on monkeys were conducted under general anesthesia following standard aseptic techniques.

Monkeys faced a Two-Alternative Forced Choice (2AFC) to guess whether an oriented drifting grating stimulus came from a narrow or wide distribution of orientations, centered on zero with standard deviations $\sigma_+ = 15°$ and $\sigma_- = 3°$. Visual contrast was set to 64%. Each trial was initiated by a beeping sound and the appearance of a fixation target (0.15° visual angle) in the center of the screen. The monkey fixated on a fixation target for 300 ms within 0.5°–1° visual angle. The stimulus appeared at the center of the screen. After 500 ms, colored targets appeared randomly on the left and right, and the monkey then saccades to one of these targets to indicate its choice (red and green targets correspond to narrow and wide distributions).

After the monkey was fully trained, we implanted a 96-electrode microelectrode array (Utah array, Blackrock Microsystems, Salt Lake City, UT, USA) with a shaft length of 1 mm over parafoveal area V1 on the right hemisphere. The neural signals were pre-amplified at the head stage by unity gain preamplifiers (HS-27, Neuralynx, Bozeman MT, USA). These signals were then digitized by 24-bit analog data acquisition cards with 30 dB onboard gain (PXI-4498, National Instruments, Austin, TX) and sampled at 32 kHz. The spike detection was performed offline according to a previously described method[12,75]. For each behavioral session and in both monkeys, 95 multiunit neural responses $r_k$ were measured by spike counts in the 500 ms preceding the saccade target onset.

The animals did not perform well on all days, so for further analysis we selected sessions where the performance exceeded 0.7 for monkey 1 (85% of all sessions) and 0.75 for monkey 2 (68% of all sessions).

The neural data from the two monkeys is of comparable quality, although the monkey with higher task accuracy (monkey 2) performs more trials and has more significantly tuned neurons. When we resampled from both datasets to control for the number of trials and tuned neurons, and then used comparable datasets to do nonlinear choice correlation analysis, we found similar decoding efficiencies for two monkeys as was reported in the section "Evidence for optimal nonlinear computation in macaque brains" (data not shown).

The task-relevant stimulus $s$ is the large or small variance $s_\pm = \sigma_\pm^2$ of the distribution over orientations. The orientation $\phi$ is a variable jointly determined by the task-relevant stimulus and a multiplicative nuisance variable $\nu$ through $\phi = \sqrt{s}\nu$, with $\nu \sim \mathcal{N}(0, 1)$. If the orientation itself can be estimated locally from linear functions of the neural responses, then the stimulus can be decoded quadratically from those neural responses according to $\hat{s} = \hat{\phi}^2$. A binary classification of the variance is given by $\hat{s}_\pm = \text{sgn}\ (\hat{\phi}^2 - \theta^2)$ where $\theta$ is the animal's orientation threshold. This threshold is optimal where the two stimuli are equally probable: $p(\phi|s_+) = p(\phi|s_-)$, implying that $\theta_{\text{opt}}^2 = (\log s_+ - \log s_-)/(s_-^{-1} - s_+^{-1})$. The probability of correctly guessing the orientation variance is $\frac{1}{2}(p(\hat{s}_\pm = +|s_+) + p(\hat{s}_\pm = -|s_-))$, where these probabilities can be computed from the cumulative normal distribution on the correct side of the optimal orientation threshold, $p(\hat{s}_\pm = +|s_+) = 2\int_{\theta_{\text{opt}}}^\infty d\phi\ p(\phi|s_+) = \text{erfc}\left(\theta_{\text{opt}}/\sqrt{2s_+}\right)$; similarly, $p(\hat{s}_\pm = -|s_-) = 1 - \text{erfc}(\theta_{\text{opt}}/\sqrt{2s_-})$. Using values of $s_\pm$ for our task, this gives an optimal fraction correct of 0.82.

We computed choice correlations using NACCC (Eq. (17)), and discriminability based on total correlations between stimulus and response (Eq. (20)). We adjusted the optimal prediction by constant factors $\zeta$ and $\delta$ to account for binary choices using the equations in Supplement S.6.4, with thresholds estimated by logistic regression between choice and the absolute value of the stimulus orientation. We estimated the slopes of the relationship between measured and predicted choice correlation using the angle of the principal component of the

bivariate data. We computed standard deviations for these quantities by bootstrapping 100 times.

For our two shuffle controls testing whether correlations between neurons were informative about the stimulus or choice, we selected responses independently from $r_i \sim p(r_i|s, \phi, \hat{s})$ (Fig. 6f) or $r_i \sim p(r_i|s, \hat{s})$ (Fig. 6g). We evaluate statistical significance of the measured and predicted optimal choice correlations using $p$-values for null distributions based on 100 shuffled choices and 100 shuffled stimuli, while preserving correlations between neural responses. Both null distributions are approximately Gaussian with zero means, so we compute the $p$-value of the choice correlations with respect to the corresponding Gaussian, $p = 1 - \frac{1}{2}\text{erfc}\left(-|x|/\sqrt{2\sigma_x}\right)$ where $x$ is the quantity of interest and $\sigma_x$ is its standard deviation (Fig. 6c, d).

**Reporting summary**. Further information on research design is available in the Nature Research Reporting Summary linked to this article.

## Data availability
The data that support the findings of this study are available from the corresponding author upon reasonable request.

## Code availability
All custom code used for electrophysiology data collection and data processing are made publicly available at github.com/atlab. Experimental data for Fig. 6 and code used for analysis and figure generation are available for download from github.com/xaqlab/nonlinear_choice_correlation.

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

## Acknowledgements

The authors thank Jeff Beck, Valentin Dragoi, Arun Parajuli, Alex Pouget, Nicole Rust, and Haim Sompolinsky for helpful conversations. This work was supported by NSF CAREER grant 1552868 to X.P., by NeuroNex grant 1707400 to X.P. and A.T., and by NSF Grant No. PHY-1748958, NIH Grant No. R25GM067110, the Gordon and Betty Moore Foundation Grant No. 2919.01. Q.Y. was supported in part by National Natural Science Foundation of China grant No. 32100832 and by the Natural Science Foundation of the Higher Education Institutions of Jiangsu Province China Grant No. 19KJD520001. X.P. and A.T. were supported in part by the Intelligence Advanced Research Projects Activity (IARPA) via Department of Interior/Interior Business Center (DoI/IBC) contract number D16PC00003. The U.S. Government is authorized to reproduce and distribute reprints for Governmental purposes notwithstanding any copyright annotation thereon. Disclaimer: the views and conclusions contained herein are those of the authors and should not be interpreted as necessarily representing the official policies or endorsements, either expressed or implied, of IARPA, DoI/IBC, or the U.S. Government.

## Author contributions

X.P. and Q.Y. conceived the theoretical framework. X.P. and Q.Y. designed and performed the mathematical analyses. Q.Y. performed the simulations. E.W., R.J.C., and A.S.T. designed the experiments for a study with Wei Ji Ma; E.W., R.J.C., and A.S.T. performed the experiments; E.W. preprocessed the neural data; Q.Y. and X.P. analyzed the neural data. Q.Y. and X.P. wrote the manuscript; all authors discussed the results and commented on the manuscript.

## Competing interests

The authors declare no competing interests.
