## [Peer Review File · Nature Communications]

Revealing nonlinear neural decoding by analyzing choicesREVIEWER COMMENTS

Reviewer #1 (Remarks to the Author):

In this work, Yang et al. take on the well-known problem of determining how much information about a stimulus is present in a population of neurons and determining how effectively the animal leverages this information in making their choice. They derive a novel metric to include nonlinear information, and demonstrate it on monkey V1 to show near-optimality. The metric is of substantial value, the derivation itself does quite a bit of work, the approach is clever, and the Discussion is fairly thorough. My only real comments regard portions of the framing.

First, some additional strengths of this manuscript. This work addresses a long-standing problem – one that is almost certainly impossible to solve for all cases – and is one of few real steps toward solving it in reasonable cases. The partitioning of noise correlations into nuisance correlations and internal correlations is insightful. The metric they derive is straightforward to use, and uses only fairly robust statistics (which is important and unusual for dealing with nonlinear measures). They demonstrate their measure on numerous synthetic cases to show that it handles some amount of model mismatch gracefully, and apply it to real data to show that monkey V1 is near-optimal with respect to a problem that can only be solved with second-order statistics. The Discussion covers a large number of important caveats. Despite the amount of math, it was a generally clear read.

My only real complaint is that critical caveats are not covered until the Discussion, and a few are missing even there. In particular, there are (unavoidably) a variety of cases where this method will fail to reveal nonlinear information. The text seems to make the claim in places that this method works for nonlinear encoding in general. It should be upfront that there are limitations to this claim. This should include the abstract.

One place this issue crops up is section 2.2.3, which argues that it's not a problem to be ignorant of the nonlinearity. Their example is impressive, but as they acknowledge later there has to be a limit. For example, the method would presumably fail for serious discontinuities (such as the Heaviside function), periodic or other nonmonotonic nonlinearities, or other functions that differ greatly from low-order polynomials.

At a greater extreme, this method is presumably only applicable to rate coding and not temporal coding. Additionally, if I understand correctly, it can only identify mappings that are $\mathbb{R}^n \rightarrow \mathbb{R}^1$. This method therefore is limited in its application to the motor decoding problem, which produces an extended and time-varying output. This is not at all a problem, the scope just needs to be clear.

Regarding the data portion, the authors find that their two monkeys have similar performance, but one has near-optimal choice decoding (implying that information is lost before V1) while the other has poor choice decoding (implying that information is lost after V1). This is definitely interesting. However, is this the only interpretation of the finding? Could something be different due to different sampling biases or having differences in SNR between the datasets? It seems like lower-quality recordings in the second animal could result in worse nonlinear decoding due to a reduced ability to infer higher-order statistics. However, the NACCC metric is subtle enough that I am not entirely clear how such differences might affect results. I would suggest that the consequences of any differences in gross properties of the datasets deserve careful thinking.

MINOR

2.2: "In principle, we could discount such indirect relationships with complete recordings of all neural activity." This doesn't seem to follow: the roles of two perfectly correlated neurons, of which only one projects to a target, can't be disentangled by recording them both. Additional information would be required.

Monkey task description: It might be helpful to note earlier that the optimal strategy is simply thresholding the deviation from a particular orientation.

First sentence of 2.3.2: It took me quite a while to understand what was meant; the coarse/fine distinction is quite field-specific and it wasn't clear why it mattered until deep in the Methods.

2.3.2: "In general, there is no guarantee that the particular nonlinear statistics that are informative about the stimulus are also informative about the choice. Our theory of optimal decoding predicts specifically that these quantities should be directly proportional to each other." Isn't this a direct consequence of the fact that the animal can do the task well, when the task requires nonlinear processing?

3.4: In the discussion of naturalistic tasks, several recent papers have made this point and could be cited. Most notable is Krakauer et al. (2017, Neuron).

Reviewer #2 (Remarks to the Author):

The manuscript proposes a nonlinear extension of the traditional encoder-decoder framework for studying how fluctuations in neural responses are linked to behavioral variability. They ask which nonlinear (point) operations seem to drive the map between the two, using data from monkey V1.

Significance:

- on the theoretical side, this work builds heavily on standard past results, with the novel dimensions being 1) the explicit segregation of noise sources into intrinsic and nuisance dimensions and 2) the nonlinearity. However, I don't find the usage of the nuisance variables as an argument for the need of nonlinear computation particularly compelling. or novel. Overall, I would assess the technical contribution as incremental.

- at the analysis level, I found the particular choice of the task strange, in that the setup seems incongruent with the setup the framework operates in (one focuses of integration of information/ sequential processing whereas the framework is single stimulus feedforward processing). Given that there seems to be a degree of alignment of the results with the predictions of the theory despite the setup not actually applying makes me doubt the overall diagnostic power of the proposed statistic as a useful tool for identifying the nature of nonlinear brain computations.

More detailed thoughts:

- I personally found the way the intro builds the setup of the theory a bit confusing, it's hard to disentangle the main point: is it about mixed selectivity, the use of marginal statistics, or nature of decoder?

- the framework of (linear gaussian) encoding-decoding models has a very long history and has served the field well in the past, but in many ways has reached the extent of its utility. What the field needs is not additional variations on the same theme, but new ideas and new ways of conceptualizing brain computation. In that sense, this work feels somewhat dated.

- same goes for the revisiting of the kernel trick.

- for the linear case, there was a lot on emphasis on the locality of the metric, which does not translate to arbitrary population level nonlinearities in eq 6; the fact that in the following sections, one can ignore the issues of unobserved subpopulations seems to imply that in practice the application of these ideas is restricted to point nonlinearities, which significantly diminishes its power.

- implicit gaussianity assumptions for response model: d' as a metric seems intrinsically limiting. Also nonlinear operations applied to gaussian variables lead to potentially drastically nongaussian statistics, which raises some additional self-consistency issues.

- i find there's a certain degree of circularity in the definition of optimality in this context, as in we

don't seem to ask which nonlinearities capture the stimulus/task statistics best but to which degree encoding and choice align for a particular nonlinearity. that seems not normative and a wrong metric of success

- a lot of the main text consists of a very expansive description of what I consider very basic well known concepts, like the linear encoding/decoding, or the kernel trick -- which would have been better spent explaining the specifics of the novel additions, in particular the computation of the nonlinear discriminability index; main text also needs a sharper conceptualization of the key steps of the approach. Seems like all of the substance is hiding in supplementaries or the methods.

- for the data part: i felt that more detail was needed, in particular spelling out how the components of the experiment map into elements in the theoretical framework, but also in the actual analysis and what the metrics being presented mean in the context of the theory.

- it would be useful to spell out more clearly what are the novel aspects, and the significance of each added element in explaining the data

Reviewer #3 (Remarks to the Author):

Yang et al presents a theoretical framework for the nonlinear decoding of sensory responses that is constrained by behavioral choices. After formalizing how to think about the role and sources of different types of variability in this process, they develop a statistic for assessing the efficiency of the decoding strategy that is employed by a brain.

The theory, which is the main focus of this manuscript, is strong and well-developed. It addresses a timely topic that is increasingly important as the complexity of experimental data expands in terms of the numbers of neurons recorded and, more relevantly, the numbers of stimulus properties studied and the degree of task complexity employed. The data described at the end of this manuscript are not a perfect fit for the theory and provide only a minor additional contribution. But, I believe that the theory and approach described in this manuscript has the potential to be of broad use to both theoretical and experimental communities.

I have a few comments that, if addressed, would strengthen the manuscript and improve its accessibility to a broad audience.

1. As this paper is aimed towards a relatively broad readership, the authors should help develop a reader's intuition for the magnitude and distribution of choice correlation measures that are used and how the response properties of measured neurons affect these values. Perhaps the strongest way to do this would be via additional analyses or simulations, but even a small amount of additional text would go a long way. As it stands, a reader is not sure what correlation values they might expect to observe in a typical dataset, how these values relate to well-established methods and how they can be expected to change based on well-described response properties.

It would help to develop intuition relating the magnitude of these choice correlation measures (the NACCC or intermediate measures) to known values and for the reader to understand what properties of neural responses their magnitude depends on. A straightforward analogy is to understanding how choice probabilities do or do not depend on a variety of measures including the amount of data, tuning properties and shared variability of recorded neurons and what can be inferred from these measures about readout (for example, pool size).

I list some example questions below, but this is not meant to be an exhaustive list to reply to. Instead, the authors should identify the aspects that are worthy of reporting with the goal of building an intuition for the reader.

For example: if shared variability in a population is higher or lower, how does that change the distribution and magnitude of these measures? What if the readout population size changes

dramatically - eg if a simulated network's decision is drawn from the activity of different sized pools of neurons, or a dramatically different weight profile, how do these distributions change. What role does the sparseness/broadness of tuning for different (relevant/nuisance) stimulus dimensions (or the rate modulation due to different stimulus features) in a population play? What are the constraints that bound these measures? Are these measures biased in some way at low trial numbers or with a response/choice bias towards one option in a 2AFC or a range of options in a continuous discrimination?

Additionally, the tools presented in this manuscript seem like they would allow for important quantitative statements about the experimental demands for their use compared to standard linear approaches. What are the experimental contexts in which the author's approach buys an experimenter additional sensitivity? Can the authors outline the tradeoffs in terms of task design (eg 2AFC vs continuous report) or stimulus set (eg the number of relevant or irrelevant stimulus dimensions that change) that best make use of their approach. For example, one may assume that a continuous discrimination task will provide additional sensitivity for understanding the mapping between visual representations and choice and that this would be even more true with this decoding approach. But, is there a cost in terms of the amount of data (#s of neurons or # of trials) needed?

2. The authors spend a fair amount of time parsing the distinction between intrinsic and extrinsic noise and the subdivision of nuisance variability. This is important, but becomes muddled by the time the data in the final figure are addressed. The motivation from the start of the paper was to draw the distinction between properties of a stimulus that change firing rates (or patterns) but are not useful for guiding a discrimination. Unfortunately, these data don't contain visual stimuli that map perfectly onto the examples that the authors developed earlier in the text and there is not a strictly irrelevant stimulus feature (like, say, phase). The authors treat the direction of the visual stimulus in the discrimination task as a nuisance variable, even though the direction of the stimulus itself is what the subject uses to determine the distribution that it originated from.

Ultimately, I do not fault the authors for this, but I think a little more care in the writing could help the reader through these issues.

Relatedly, the authors state:

"Combining the conclusions from these controls, we find no evidence that the brain optimally decodes any stimulus-dependent internal noise correlations in this task."

Have the authors demonstrated that there actually are any stimulus-dependent internal noise correlations in this task? Do the authors simply mean that the distributions have changed between the shuffle analyses in E, F and G of Figure 6? Can they please clarify?

3. Finally, this topic may be well beyond the scope of the current manuscript, but it will immediately spring to a reader's mind. In the discussion, the authors note the potential limitations of this approach for addressing complex IT responses. But what about intermediate steps in visual processing? I'm curious to know whether the authors see this approach as being useful for understanding the transformation and construction of distinct visual features from subordinate features and how those responses are used to guide behavior.

The concept of nuisance correlations seems to make more sense for an animal engaged in a task than it does for neurons within any particular visual brain area. Further, what is/is not a nuisance may differ across (or even within) visual areas and one neuron's nuisance variable may well be another neuron's signal.

For example, the authors use phase as an example in the text and in Figure 2. Similarly, they could have used spatial frequency, which is useful for the following, more complicated, example. If these Gabors drifted, their SF, drift rate (phase velocity) and orientation would be used to construct an estimate of the stimulus velocity. These features tend to be represented somewhat separably in V1 simple cells (eg, see Mazer et al, 2002; Priebe et al 2006) but are combined by complex cells in V1 or MT neurons to represent stimulus speed.

Does this slight elaboration break down the tools developed here (as the authors suggest might

happen in IT), or are they robust to this example? Could the magnitude of choice correlations in different populations of neurons provide insight into how a choice is formed based on a 'constructed' visual property like speed? Does this example pose a problem for the current definition of 'nuisance' variability that the authors have employed?

Remarks from the reviewers are shown in black.

Our responses are shown in blue.

Original text is quoted in green.

Updated text is quoted in red, and marked in red in the new manuscript.

Many thanks for your further consideration.

Best,

Xaq Pitkow

REVIEWER COMMENTS

Reviewer #1 (Remarks to the Author):

In this work, Yang et al. take on the well-known problem of determining how much information about a stimulus is present in a population of neurons and determining how effectively the animal leverages this information in making their choice. They derive a novel metric to include nonlinear information, and demonstrate it on monkey V1 to show near-optimality. The metric is of substantial value, the derivation itself does quite a bit of work, the approach is clever, and the Discussion is fairly thorough. My only real comments regard portions of the framing.

First, some additional strengths of this manuscript. This work addresses a long-standing problem – one that is almost certainly impossible to solve for all cases – and is one of few real steps toward solving it in reasonable cases. The partitioning of noise correlations into nuisance correlations and internal correlations is insightful. The metric they derive is straightforward to use, and uses only fairly robust statistics (which is important and unusual for dealing with nonlinear measures). They demonstrate their measure on numerous synthetic cases to show that it handles some amount of model mismatch gracefully, and apply it to real data to show that monkey V1 is near-optimal with respect to a problem that can only be solved with second-order statistics. The Discussion covers a large number of important caveats. Despite the amount of math, it was a generally clear read.

We thank the reviewer for the appreciative comments.

My only real complaint is that critical caveats are not covered until the Discussion, and a few are missing even there. In particular, there are (unavoidably) a variety of cases where this method will fail to reveal nonlinear information. The text seems to make the claim in places that this method works for nonlinear encoding in general. It should be upfront that there are limitations to this claim. This should include the abstract.

We now summarize these caveats in the abstract, as:

“This relationship holds for optimal feedforward networks of modest complexity, when experiments are performed under natural nuisance variation.”

There are considerably more challenges when not just testing for optimality but when trying to identify the particular suboptimality, but these will wait for our next paper.

One place this issue crops up is section 2.2.3, which argues that it’s not a problem to be ignorant of the nonlinearity. Their example is impressive, but as they acknowledge later there has to be a limit. For example, the method would presumably fail for serious discontinuities (such as the Heaviside function), periodic or other nonmonotonic nonlinearities, or other functions that differ greatly from low-order polynomials.

Although we used polynomials here, there is no fundamental requirement to use that nonlinear basis. In the original text in section 2.2.5 (We have moved previous section 2.2.3 to section 2.2.5), we say: “The prediction is valid as long as the brain’s nonlinearity can be expressed as a linear combination of the tested nonlinearities (Methods 4.2.2).”

It is true that when we approximate complicated or sharp nonlinearities, we may need many more polynomial basis components compared to the number for evaluating simpler nonlinearities, or use a basis including such complexities or discontinuities. We now acknowledge this directly in section 2.2.5:

“Since the brain needs complicated nonlinearities for complicated tasks, it may be difficult to find a suitable basis set for truly natural conditions; feature spaces from deep networks trained on comparable tasks might provide a useful basis. For the modest, controlled-complexity tasks used in most neuroscience experiments, polynomials or other

simple bases may be sufficient, even when individual neurons do not use polynomial operations.”

At a greater extreme, this method is presumably only applicable to rate coding and not temporal coding.

Actually, our method is applicable to temporal coding as well. Instead of using spike count as the considered statistics, one could instead consider temporal statistics such as time to first spike after the stimulus onset, characteristics based on the second and higher statistical moments of the ISI probability distribution, spike jitter, etc. One could then ask how much information those statistics contain about a stimulus or behavioral choice, and apply our nonlinear choice correlation test based on these temporal neural statistics. We now mention this in the Discussion:

“Our theory describes choice correlations for a snapshot of neural activity and a static, scalar choice variable. However, our method can also be applied to temporal neural response properties like spike timing or time series. For optimal processing, spike timing that is tuned to task-relevant stimuli should also be correlated with resultant choices, even if the timing is converted to rate codes by downstream processing.”

Additionally, if I understand correctly, it can only identify mappings that are $\mathbb{R}^n \rightarrow \mathbb{R}^1$. This method therefore is limited in its application to the motor decoding problem, which produces an extended and time-varying output. This is not at all a problem, the scope just needs to be clear.

This is an interesting topic. In this paper we indeed addressed scalar outputs. But one could also consider multivariate outputs, like open-loop motor tracking of a target in multiple dimensions or a scalar over time, and then assess how different neural statistics affect each dimension of the multidimensional output. The covariance of an unbiased multivariate estimator is bounded by the Fisher Information Matrix, just as the variance is bounded by the Fisher Information (scalar). The matrix bound $J^{-1} \leq \Sigma$ means that $\Sigma - J^{-1}$ is positive semidefinite, *i.e.* the estimator variance in any dimension is bounded by the Fisher information in that dimension. This means that we can treat each behavioral dimension (over time or space) as having its own statistical relationship with the neural fluctuations, and our test holds for each: the neural information about any stimulus pattern should be proportional to the neural information about a corresponding behavioral estimator. That allows us to consider $\mathbb{R}^n \rightarrow \mathbb{R}^{k \times T}$ in a feedforward setting. That said, *recurrent* temporal structure as used in motor control is indeed more complicated, so we do not elaborate here but mention the challenge in the Discussion after talking about the spike timing information:

“On the other hand, it is more difficult to track behavioral consequences of spatiotemporal correlations that evolve through a recurrent network (Lakshminarasimhan *et al* 2018) with dynamic outputs, as in motor control applications. It should be fruitful to develop this theory further for more complex decoding tasks.”

Regarding the data portion, the authors find that their two monkeys have similar performance, but one has near-optimal choice decoding (implying that information is lost before V1) while the

other has poor choice decoding (implying that information is lost after V1). This is definitely interesting. However, is this the only interpretation of the finding? Could something be different due to different sampling biases or having differences in SNR between the datasets? It seems like lower-quality recordings in the second animal could result in worse nonlinear decoding due to a reduced ability to infer higher-order statistics. However, the NACCC metric is subtle enough that I am not entirely clear how such differences might affect results. I would suggest that the consequences of any differences in gross properties of the datasets deserve careful thinking.

The neural data from the two monkeys is of comparable quality and taken from the same visual eccentricities, although the better monkey also performs more trials and has more significantly tuned neurons (Figure R1). On average, monkey 2 performed more trials in each session (752) than monkey 1 (613 trials), and had more significantly tuned neurons (m2:60, m1:49). To rule out the possible influence of data quality difference onto our conclusions, we resample from both datasets to control for their numbers of trials and tuned neurons used for choice correlation analysis (Figure R1B). On average, in each session, monkey 2's new dataset contains 607 trials and 43 neurons, where monkey 2's dataset contains 602 trials and 47 neurons. The choice correlation tests give the same conclusion about their decoding efficiency (Figure R1C,D,E; compared with the conclusion in the main text (Figure 6E,F,G). Monkey 1 has a decoding efficiency of 0.76 ± 0.05 (mean \pm 95% confidence intervals). Monkey 2 has a decoding efficiency of 0.99 ± 0.03 . These controls are consistent with our conclusions about the two monkeys' decoding efficiency in Section 2.3. It will be interesting in future work to explore hypotheses about the cause of the suboptimal decoding efficiency, and to identify corroborating evidence for our finding that the monkeys use strategies with different decoding efficiencies.

Figure R1: Data quality for the two monkeys used in our study. **A. top:** Histogram over sessions of the number of trials. Monkey 2 generally performed more trials than monkey 1. **bottom:** Histogram over sessions of the number of neurons significantly tuned to orientation. Significantly tuned is defined as neurons whose correlation with orientation has a p -value less than 0.01. Our data revealed more significantly tuned neurons for monkey 2 than for monkey 1. **B.** Controlled datasets were generated by subsampling Monkey 2's recorded neurons and trials so that both monkeys have the same data quantity. **C.** Nonlinear choice correlation tests with these resampled neural datasets. The measured and optimal NACCCs are still highly correlated ($r = 0.86, 0.71, 0.78$ for linear, square and cross terms for monkey 1; $0.83, 0.85, 0.83$

monkey 2). **D.** Shuffling internal noise correlations while preserving nuisance correlations maintains the relationship between prediction and nonlinear choice correlations, implying that internal noise is not responsible for the correlations. **E.** Shuffling nuisance correlations across trials nearly eliminates the relationship between measured and predicted nonlinear choice correlations (0.86, 0.05, 0.05 for monkey 1; 0.83, 0.19, 0.18 for monkey 2), implying that nuisance variation creates the nonlinear code.

We now mention this in Methods Section 4.4:

“The neural data from the two monkeys is of comparable quality and visual eccentricity, although the monkey with higher task accuracy (monkey 2) performs more trials and has more significantly tuned neurons. When we resampled from both datasets to control for the number of trials and tuned neurons, and then used comparable datasets to do nonlinear choice correlation analysis, we found similar decoding efficiencies for two monkeys reported in Section 2.3 (data not shown).”

MINOR

2.2: “In principle, we could discount such indirect relationships with complete recordings of all neural activity.” This doesn’t seem to follow: the roles of two perfectly correlated neurons, of which only one projects to a target, can’t be disentangled by recording them both. Additional information would be required.

True. But neurons are never *perfectly* correlated. Even so, with finite data, it can be difficult to identify the true causal influence, as we mentioned in the next line: In principle, we could discount such indirect relationships with complete recordings of all neural activity. This is currently impractical for most animals, and even if we could record from all neurons simultaneously, we would struggle to acquire enough trials to fully disambiguate how neural activities directly influence behavior.

To further emphasize this point, we replace “complete recordings” with “exhaustive recordings”.

Monkey task description: It might be helpful to note earlier that an optimal strategy is simply thresholding the deviation from a particular orientation.

We now expand Section 2.3.1 where we describe the task and its relevant and irrelevant variables. We now say “Note that for a fixed contrast, an optimal computation based on the stimulus is simply to threshold the squared deviation from the mean orientation.”

First sentence of 2.3.2: It took me quite a while to understand what was meant; the coarse/fine distinction is quite field-specific and it wasn’t clear why it mattered until deep in the Methods.

Apologies for this oversight, and we thank the reviewer for their persistence in trying to figure this out. We now explain this more fully, earlier in the text in Section 2.2.7, Coarse versus fine, estimation versus classification:

“Our conceptual framework and predictions are most simply expressed for fine continuous estimation tasks, where here we define ‘fine’ as a stimulus range over which the noise statistics do not vary with the stimulus. Some minor details of our predictions change when moving to binary classification instead of continuous estimation: this introduces a correction factor that depends on the response distribution (Section S.6.4).

More details change when moving to ‘coarse’ tasks, which we define as when noise statistics do change significantly with the stimulus. As for fine discrimination, we again find that when decoding is optimal, random fluctuations in choices are correlated with neural responses to the same degree that those responses can discriminate between stimuli. However, this relationship is slightly more complicated for coarse discrimination. For this reason we introduce a slightly more complicated measure of choice correlation that we call Normalized Average Conditional Choice Correlation (NACCC, Eq. 17). However, the end result is the same: choice correlations for optimal decoding are equal to the ratio of discriminabilities (Eq. 7; Supplemental Information S.5). As for fine estimation, there is a correction factor of order 1 for binary choices instead of continuous estimation (Methods 4.2.1, Supplemental Information S.6.4, Eq. 185). In Section 2.3, when we apply our choice correlation test to real neural data, we account for both of these refinements.”

2.3.2: “In general, there is no guarantee that the particular nonlinear statistics that are informative about the stimulus are also informative about the choice. Our theory of optimal decoding predicts specifically that these quantities should be directly proportional to each other.” Isn’t this a direct consequence of the fact that the animal can do the task well, when the task requires nonlinear processing?

No: the animal could behave suboptimally even while performing the task “well”, and the reason for the suboptimality could be poor encoding or poor decoding (or both). If there is suboptimal encoding but optimal decoding, then we show that this proportionality holds. If there is optimal encoding but suboptimal decoding, then the proportionality does not hold. And animals in these two different cases could have the same performance. This is in fact what we find with our monkeys in Section 2.3.

3.4: In the discussion of naturalistic tasks, several recent papers have made this point and could be cited. Most notable is Krakauer et al. (2017, Neuron).

Now cited in the discussion, along with Niv (2020).

Reviewer #2 (Remarks to the Author):

The manuscript proposes a nonlinear extension of the traditional encoder-decoder framework for studying how fluctuations in neural responses are linked to behavioral variability. They ask which nonlinear (point) operations seem to drive the map between the two, using data from monkey V1.

Significance:

- on the theoretical side, this work builds heavily on standard past results, with the novel dimensions being 1) the explicit segregation of noise sources into intrinsic and nuisance dimensions and 2) the nonlinearity. However, I don't find the usage of the nuisance variables as an argument for the need of nonlinear computation particularly compelling, or novel. Overall, I would assess the technical contribution as incremental.

The reviewer is correct that it is possible to have *linear* combinations of task variables and nuisance variables. In fact we address this extensively in Section 3.3 and Section S4, which shows how the choice correlation structure is influenced by the presence or absence of natural nuisance variation.

However, natural nuisance variation is not linear and additive: that is by far the exception rather than the rule. And typically, it is these nonlinear interactions that entangle the task variables in the sensory domain. Consider color constancy, shape from shading, auditory localization, olfactory recognition: all of these are hard problems precisely because of nonlinear interactions with nuisance variables (e.g. illuminant, intensity). Indeed, we're struggling to understand the reviewer's juxtaposition of "not compelling" and "not novel." If it is not novel, then surely that is because past literature has acknowledged that entanglement by nuisance variation is a compelling problem (DiCarlo and Cox 2007, Pinto et al 2008, Beck et al 2012, Anselmi et al 2020).

For the technical contributions, we hope the reviewer appreciates the importance and difficulty of getting the details right. We are dealing with nonlinear computation and non-Gaussian statistics, and finding the appropriate relationships for quantitative experimental predictions is nontrivial compared to the linear case (see particularly Sections S3–S6).

- at the analysis level, I found the particular choice of the task strange, in that the setup seems incongruent with the setup the framework operates in (one focuses on integration of information/ sequential processing whereas the framework is single stimulus feedforward processing). Given that there seems to be a degree of alignment of the results with the predictions of the theory despite the setup not actually applying makes me doubt the overall diagnostic power of the proposed statistic as a useful tool for identifying the nature of nonlinear brain computations.

We are a little confused about the concern here. The tasks we analyze are essentially static estimation and classification tasks, counting spikes in a time bin during presentation of a drifting grating, and we assume a feedforward processing chain for the brain to solve this task.

None of our tasks, either as analyzed or performed by the monkey, require temporally sequential processing. We state in the Discussion that:

“Our theory describes choice correlations for a snapshot of neural activity and a static, scalar choice variable. However, our method can also be applied to temporal neural response properties like spike timing or time series. For optimal processing, spike timing that is tuned to task-relevant stimuli should also be correlated with resultant choices, even if the timing is converted to rate codes by downstream processing.”

More detailed thoughts:

- I personally found the way the intro builds the setup of the theory a bit confusing, it's hard to disentangle the main point: is it about mixed selectivity, the use of marginal statistics, or nature of decoder?

Our main point is the connection of all three ideas: we analyze how marginal statistics are decoded from neurons of mixed selectivity. We now clarify this with the following text in the Introduction:

“This paper makes four main contributions. First, it weaves together important concepts about tuning curves and nuisance variables, nonlinear computation, and redundant population codes, forming a general, unified description of feedforward encoding and decoding processes in the brain. This description is supported by intuitive explanations and concrete examples to illustrate how these concepts relate to each other and enrich familiar views of neural computation. Second, this paper provides a simple way of testing the hypothesis that the brain's decoding strategy is efficient, using a simple, novel statistic to assess whether neural response patterns that are informative about the task-relevant sensory input are also informative about the animal's behavior in the task. Third, it establishes the technical details needed to apply this test in practical neuroscience experiments. Fourth, we apply this test to analyze V1 data from macaque monkeys, finding direct experimental evidence for optimal nonlinear decoding.”

- the framework of (linear gaussian) encoding-decoding models has a very long history and has served the field well in the past, but in many ways has reached the extent of its utility. What the field needs is not additional variations on the same theme, but new ideas and new ways of conceptualizing brain computation. In that sense, this work feels somewhat dated.

We agree that the framework of linear gaussian encoding-decoding models is fundamentally limiting, but we did *not* merely use linear gaussian encoding-decoding models. Indeed, that is the central purpose of our study. In our feedforward chain, we extend our encoding model to allow members of the exponential family with nonlinear sufficient statistics. This includes far more options than merely linear gaussian. Indeed, coming up with models for cubic sufficient statistics to illustrate the non-linearity and non-gaussianity was quite complicated (Methods 4.1.4, Supplement S.1.4), precisely because it falls outside of the usual mathematical tools. Moreover, we do consider the implications of flexible nonlinear neural networks to extract the information as well (Figure 4), and we discuss how our method applies to the statistics

computed by arbitrary nonlinear network transformations (Results 2.2.5, Discussion 3.1, Methods 4.2.2).

- same goes for the revisiting of the kernel trick.

We don't actually use the kernel trick, which avoids explicit high-dimensional embeddings by transforming the *distances* between data points. Here we actually use high-dimensional nonlinear representations directly.

But we think the reviewer's real concern here is that shallow nonlinear computation for untangling is dated. There's a good reason why nonlinear computation is a core ingredient in all models of neural processing, and that is because it is required to perform natural tasks. Whether the network is shallow or deep, feedforward or recurrent, is tangential to our central point. When nonlinear transformations get too complicated, then our approach will not be useful, but for many neuroscience tasks our simple approach can be helpful, and more illuminating than black-box deep networks. We state in the Abstract,

"This relationship holds for optimal feedforward networks of modest complexity, when experiments are performed under natural nuisance variation."

In section 2.2.5 we elaborate:

"Since the brain needs complicated nonlinearities for complicated tasks, it may be difficult to find a suitable basis set under truly natural conditions; feature spaces from deep networks trained on comparable tasks might provide a useful basis. For the modest, controlled-complexity tasks used in most neuroscience experiments, polynomials or other simple bases may be sufficient, even when individual neurons do not use polynomial operations."

Moreover, our paper doesn't merely revisit nonlinear computation as a way to solve nonlinear untangling, but additionally relates the well-established idea of nonlinear untangling to higher-order sensory statistics, and the propagation of those statistics to choices — and builds intuitions about this process through practical examples.

Finally, we provide a new test to analyze nonlinear neural computations all the way from the sensory input to the decision-making output, a critical tool needed with the advent of modern tools like NeuroPixels and multiple Utah arrays that allow larger population recordings simultaneously with behavior. It can be readily applied to common tasks in neuroscience and meet the needs of current large-scale neural recording analysis.

- for the linear case, there was a lot on emphasis on the locality of the metric, which does not translate to arbitrary population level nonlinearities in eq 6; the fact that in the following sections, one can ignore the issues of unobserved subpopulations seems to imply that in practice the application of these ideas is restricted to point nonlinearities, which significantly diminishes its power.

Perhaps the reviewer has misunderstood our concept of 'locality' here? The locality of our metric is with respect to the stimulus, not with respect to neurons, per usual for the Fisher Information. Because several of our proposed tasks are fine estimations about a stimulus near a

reference, this requires the brain to do 'local' decoding based on the neural activity when it receives similar stimulus input. We use fine estimation only because it is the simplest case and extremely common in neuroscience; we also elaborate extensively on nonlocal coarse tasks as well, and develop and test mathematical predictions for those as well.

In contrast, the 'locality' in the reviewer's mind seems to be with respect to neural populations (although we may be incorrect about what the reviewer has in mind). We didn't assume that the brain only decodes 'local' subpopulation of neurons. Instead, in both our Quadratic codes, Cubic codes and application to experiment sessions, we consider the brain is doing fully distributed nonlinear population decoding — e.g. decoding arbitrary pairs and triples of neurons, or even higher-order combinations — not just point nonlinearities. Figure 4 even shows how an unobserved multilayer fully-connected neural network, with an arbitrary feedforward population level nonlinearity, conforms to our predictions.

We now describe this locality more verbosely in Section 2.2.7, as also described in response to R1's question about coarse versus fine estimation:

“Our conceptual framework and predictions are most simply expressed for fine continuous estimation tasks, where here we define 'fine' as a stimulus range over which the noise statistics do not vary with the stimulus. Some minor details of our predictions change when moving to binary classification instead of continuous estimation: this introduces a correction factor that depends on the response distribution (Section S.6.4).

More details change when moving to 'coarse' tasks, which we define as when noise statistics change significantly with the stimulus. As for fine discrimination, we again find that when decoding is optimal, random fluctuations in choices are correlated with neural responses to the same degree that those responses can discriminate between stimuli. However, this relationship is slightly more complicated for coarse discrimination. For this reason we introduce a slightly more complicated measure of choice correlation that we call Normalized Average Conditional Choice Correlation (NACCC, Eq. 17). However, the end result is the same: choice correlations for optimal decoding are equal to the ratio of discriminabilities (Eq. 7; Supplemental Information S.5). As for fine estimation, there is a correction factor of order 1 for binary choices instead of continuous estimation (Methods 4.2.1, Supplemental Information S.6.4, Eq. 185). In Section 2.3, when we apply our choice correlation test to real neural data, we account for both of these refinements.”

- implicit gaussianity assumptions for response model: d' as a metric seems intrinsically limiting. Also nonlinear operations applied to gaussian variables lead to potentially drastically nongaussian statistics, which raises some additional self-consistency issues.

The reviewer is correct that d' is intrinsically limiting, but not because responses are non-gaussian. For *fine* estimation, the estimator can be well described by d' even when the response distribution is highly non-gaussian. This is because the discriminability d' reflects *stimulus* dependence of $p(r|s)$, not the *response* dependence of that probability. This can be understood by seeing that (1) d' is bounded by the Fisher Information $J(s)$ through $d' \leq ds\sqrt{J(s)}$ (Series, Stocker, Simoncelli 2009), and (2) the *observed* Fisher information is defined for a *fixed*

response, and this information is merely averaged over responses to get the (*expected*) Fisher information $J(s)$.

A real limitation is that d' only captures the linear portion of the response information (Beck et al. 2011), namely, that contained in changes in the mean response compared to the standard deviation. However, we avoid the limitations of d' by including nonlinear functions of the raw non-gaussian response, specifically to transfer information from higher-order response statistics into means. The tuning of these nonlinear functions thereby captures the information missed by d' applied to the raw responses alone. Table 1 and associated text in Section 2.1.4 emphasizes this point: “Unlike a linear code, the information is not encoded in mean neural responses but instead by higher-order statistics of responses. These functional and statistical views are naturally linked because estimating higher-order statistics requires nonlinear operations. One analytically straightforward example of a non-linear code is a stimulus-dependent covariance $Q(s) = \langle rr' | s \rangle$; its information can be decoded by quadratic operations $R = rr'$. Table 1 compares the relevant neural response properties for linear and nonlinear codes and this quadratic example.” The non-gaussian information in response variables is then quantified by $d'(R_k(r))$.

For coarse measurements, discriminability can be expressed as an integral of Fisher Information even for non-gaussian response variables, through $d' = \int_{s_-}^{s_+} ds \sqrt{J(s)}$. However, the theory and predictions proved simpler using the standard binary d' . There may be some more general reformulations of our findings in a different mathematical language, perhaps using information theory, and this is something we plan to investigate in the future.

- i find there's a certain degree of circularity in the definition of optimality in this context, as in we don't seem to ask which nonlinearities capture the stimulus/task statistics best but to which degree encoding and choice align for a particular nonlinearity. that seems not normative and a wrong metric of success

It seems the reviewer's intuition is built on the more common approach of evaluating the quality of **encoding** — a well-studied problem asking which nonlinearities capture the stimulus or task statistics. In contrast, our paper is dedicated to understanding the quality of **decoding** — how well the brain uses whatever information it has at a given stage, regardless of whether that information resulted from a good or bad sensing of the environment. As we say in the Introduction, “we must understand how encoding and decoding are related, i.e. how the brain uses the information it has. These are distinct processes, so the brain could encode a stimulus well while decoding it poorly, or vice-versa.”

Normative claims are always based on some constraints on the available data. Most normative approaches in neuroscience start from the task itself, not from the neuronal responses. While we quantify the encoding quality through tuning and information, we do not compare to optimal encoding given the proximal sensory evidence (e.g. retinal input). Here our approach begins instead with the neural data, and provides a normative model starting from that information.

- a lot of the main text consists of a very expansive description of what I consider very basic well known concepts, like the linear encoding/decoding, or the kernel trick -- which would have been better spent explaining the specifics of the novel additions, in particular the computation of the nonlinear discriminability index; main text also needs a sharper conceptualization of the key steps of the approach. Seems like all of the substance is hiding in supplementaries or the methods.

Our points on these topics are actually surprisingly subtle, and we think they warrant an exposition that is concise but also precise and intuitive since they are the ingredients to be connected in the paper. We expect that many readers and users of our approach may also be less familiar with these ideas than expert reviewers. These include the distinction between optimal *encoding* and optimal *decoding* (see preceding answer), the role of external nuisance variables versus internal noise correlations, and the statistical perspective on what nonlinearities extract. We also provide concrete examples for V1 complex cells and XOR tasks to help connect abstract concepts to examples familiar to neuroscientists. It is our experience that many readers appreciate the conceptual clarity that helps situate our contributions and tie together several threads.

In contrast, we view the nonlinear discriminability index as a technical detail — a way of implementing the core concepts. To prevent the targeted neuroscience readership from missing the conceptual forest for the mathematical trees, we relegated the core derivations to the methods and the technical details to the supplement for the interested reader.

However, we do agree that the main text benefits from a sharper conceptualization of the key steps. At the end of the introduction, we now provide the reader with a more explicit roadmap that should help frame the topics, in addition to defining the paper's primary contributions as described above in response to this reviewer's first "more detailed thoughts":

"The Results section describes the main concepts, their formal connections, and applications. Section 2.1 introduces a framework for understanding nonlinear computation, including basic notation, the internal and external (nuisance) noise and their effects on signal, and how information can be isolated by nonlinear computation of the right statistics. Section 2.2 introduces a formalism for decoding, the notion of linear and nonlinear choice correlations, fine and coarse estimation tasks, and predictions about those correlations under optimal decoding. This section continues by describing how redundancy in the population responses appears as special high-order response statistics, and how they affect the predictions. Section 2.3 presents an experimental application of these ideas. A sketch of the details of our general predictions are presented in the Methods (4.2), and are derived in full in the Supplement along with details of their application to specific models and our experimental data."

- for the data part: i felt that more detail was needed, in particular spelling out how the components of the experiment map into elements in the theoretical framework, but also in the actual analysis and what the metrics being presented mean in the context of the theory.

We now explain the match between experimental and theoretical / analysis components at the beginning of Section 2.3.1:

“This coarse binary discrimination task is a simplified version of a task that might arise in nature when identifying a surface texture or material [58]; the orientation of the material would be a nuisance variable independent of the material type. Here the observable variable, the orientation, is the product of the target variable and a nuisance variable ν . An additional nuisance variable was the stimulus contrast varying independently of the stimulus variance, although here we only analyze the highest contrast.

Below we analyze whether the trial-by-trial nonlinear statistics of V1 multi-unit neural responses to these stimuli provide information about the task-relevant category and the behavioral choice, and whether these two informations are correlated as predicted by our optimal decoding theory. Since the marginal response statistics depend substantially on the stimulus category, at least in part because the nuisance distributions differ, this is a coarse binary discrimination task. For this reason we test suitable optimal decoding predictions about choice correlations in coarse tasks, using the NACCC measure that we outlined in Section 2.2.3 and derived in 4.4 and Supplemental Information S.6.”

- it would be useful to spell out more clearly what are the novel aspects, and the significance of each added element in explaining the data

The beginning of the discussion lays out the novel aspects, and we now provide some additional guideposts to the reader:

“This study introduced a theory of nonlinear population codes, grounded in the natural computational task of separating relevant and irrelevant variables. The theory considers both encoding and decoding — how stimuli drive neurons, and how neurons drive behavioral choices. It showed how correlated fluctuations between neural activity and behavioral choices could reveal the efficiency of the brain's decoding. Unlike previous theories of nonlinear population codes [2,27], ours remains consistent with biological constraints due to the large cortical expansion of sensory representations by incorporating redundancy through a nonlinear generalization of information-limiting correlations [3]. Also unlike past work which largely concentrates on encoding efficiency, or at most on *linear* decoding only [26,46], we provide mathematical methods to quantify nonlinear decoding efficiency. Crucially, this theory provides a new and remarkably simple test to determine if downstream nonlinear computation decodes all that is encoded. When we applied this novel test to the neural responses of monkeys performing a discrimination task in which neural statistics were dominated by nuisance variation, we found quantitative results consistent with efficient nonlinear decoding of V1 activity.”

Reviewer #3 (Remarks to the Author):

Yang et al presents a theoretical framework for the nonlinear decoding of sensory responses that is constrained by behavioral choices. After formalizing how to think about the role and

sources of different types of variability in this process, they develop a statistic for assessing the efficiency of the decoding strategy that is employed by a brain.

The theory, which is the main focus of this manuscript, is strong and well-developed. It addresses a timely topic that is increasingly important as the complexity of experimental data expands in terms of the numbers of neurons recorded and, more relevantly, the numbers of stimulus properties studied and the degree of task complexity employed. The data described at the end of this manuscript are not a perfect fit for the theory and provide only a minor additional contribution. But, I believe that the theory and approach described in this manuscript has the potential to be of broad use to both theoretical and experimental communities.

I have a few comments that, if addressed, would strengthen the manuscript and improve its accessibility to a broad audience.

1. As this paper is aimed towards a relatively broad readership, the authors should help develop a reader's intuition for the magnitude and distribution of choice correlation measures that are used and how the response properties of measured neurons affect these values. Perhaps the strongest way to do this would be via additional analyses or simulations, but even a small amount of additional text would go a long way. As it stands, a reader is not sure what correlation values they might expect to observe in a typical dataset, how these values relate to well-established methods and how they can be expected to change based on well-described response properties.

It would help to develop intuition relating the magnitude of these choice correlation measures (the NACCC or intermediate measures) to known values and for the reader to understand what properties of neural responses their magnitude depends on. A straightforward analogy is to understand how choice probabilities do or do not depend on a variety of measures including the amount of data, tuning properties and shared variability of recorded neurons and what can be inferred from these measures about readout (for example, pool size).

I list some example questions below, but this is not meant to be an exhaustive list to reply to. Instead, the authors should identify the aspects that are worthy or reporting with the goal of building an intuition for the reader.

These are worthwhile questions to consider, and we appreciate the thought the reviewer has put into listing them. Many do not have a well-defined answer since they depend on the task, neural population, and decoding performance. An important contribution of our study is to identify relationships between choice correlations and tuning that hold for optimal decoding and yet do *not* depend on the task or population details. Consistent with this emphasis of the paper, below we answer these questions *specifically under the assumption of optimal decoding*, where we can make clear predictions and provide concrete intuitions.

For example: if shared variability in a population is higher or lower, how does that change the distribution and magnitude of these measures?

What if the readout population size changes dramatically - eg if a simulated network's decision is drawn from the activity of different sized pools of neurons, or a dramatically different weight profile, how do these distributions change.

Shared variability is a multidimensional quantity reflected in the response covariance and other higher-order joint statistics. Only information-limiting dimensions will affect the behavior when decoding is optimal, as the decoder is orthogonal to the remaining coordinates. When this information-limiting component of shared variability increases while the individual variability remains constant, this will decrease the population information content and increase the choice correlations. Conversely, if the individual variability increases while the information-limiting component remains constant, then the choice correlations are diluted by the additional non-information-limiting (and non-decoded) noise. To maintain the prediction for optimal decoding, the individual statistics' information content decreases by the same amount as the choice correlation.

The NACCC depends on the readout population size through the information content of those neurons. A large redundant population can have less information than a small non-redundant population, and for optimal decoding it is only the fraction of information contained within a single statistic that determines its nonlinear choice correlation. If the readout population grows and the information grows in tandem, then the choice correlations drop. If the readout population grows and information stays constant (say, because the sensory population is fixed), then the choice correlations stay constant too.

The scaling with population size seems more worrisome when we consider higher-order nonlinearities as the task-relevant statistics, since the number of raw statistics grows with the order of the nonlinearity. Cubic terms like $r_i r_j r_k$ scale in number as N^3 for N neurons in the decoded population. Thankfully, the scaling with the number of neurons depends crucially on the amount of information-limiting noise. With $M \ll N$ neurons worth of sensory information, due to information-limiting noise, then one only needs $M^3 \ll N^3$ worth of cubic neuronal statistics. We introduce a new Supplementary Information section S.7 and Figure S5A to explain this. Figure S5B also shows that the choice correlation test remains consistent with optimal decoding even when the number of recorded statistics severely undersamples the total number of informative statistics in an optimally decoded population. However, one must still heed the warnings discussed in Section 3.2: **If the brain neglects some of the informative sufficient statistics, and we don't test these neglected statistics either, then we could find the brain is consistent with our optimal decoding test, yet still be suboptimal.**

We now explain these consequences in Section 2.2.4 when we have finished introducing nonlinear choice correlations and redundant codes:

"The simple formula of Eq. 7 provides useful insights into the relationship between neural activity and choice when that activity is decoded optimally in a natural task. First, the choice correlations do not depend on the shape or magnitude of the internal noise or nuisance correlations, because these dimensions are deliberately avoided by optimal decoding, whose weights cancel those correlations (Eq. 18). The only aspect of shared variability that matters for choice is the information-limiting component, i.e. that which is indistinguishable from a change in the task-relevant stimulus. This information-limiting variability increases choice correlations mostly by decreasing the overall behavioral discriminability d' ; the shared variability is large only

at the population level and typically only has a small contribution to any one statistic [3, 28] and thus to its discriminability d'_k . With a smaller denominator and roughly unaffected numerator, the ratio in Eq. 7 rises with information-limiting correlations.

Likewise, the total number of the decoded statistics only affects the prediction of Eq. 7 insofar as it affects the available information. This is especially important when considering nonlinear statistics, as there are potentially so many of them. When optimally decoded, greater numbers of independently informative statistics will increase the total discriminability exhibited by the behavioral choices, while the discriminability for each statistic remains unchanged. According to Eq. 7, choice correlations for optimal decoding are the ratio of these two, so as the behavioral choice d' increases and the individual terms remain fixed, the choice correlations shrink. On the other hand, greater numbers of *redundant* statistics change neither the total information content nor the behavioral choice. These added statistics can have similar tuning and similar fluctuations as each other (which is what makes them redundant). Any single redundant statistic might or might not be decoded, but it is correlated with others that are. According to Eq. 7, the individual d'_{R_k} are unchanged when adding more redundant statistics; the total information d' is unchanged; and thus their ratio is fixed, consistent with the unchanged choice correlations.

When there are many fewer sensory inputs than cortical neurons, as seen in the brain, many distinct statistics $R_k(r)$ will carry redundant information. Under these conditions, many choice correlations C_{R_k} can be quite large even for optimal nonlinear decoding: the discriminabilities d'_{R_k} of redundant statistics can be comparable to the discriminability d' of the whole population, producing ratios d'_{R_k}/d' that are a significant fraction of 1 (Figure 5). Supplementary Information S.7.1 illustrates this effect for a redundant nonlinear population code: the brain need not decode all functions of all neurons to extract essentially all of the information (Figure S6A), and neuroscientists need not compute choice correlations for all possible statistics to establish decoding efficiency (Figure S6B). ”

...or a dramatically different weight profile, how do these distributions change.

Weight profiles are dictated by the non-information-limiting noise that must be avoided for optimal decoding, according to Eq 16. In principle, there may be a range of equally good readout weights. For example, given two perfectly redundant neurons, an optimal readout can select neuron 1 or neuron 2 or any weighted average of both, and they'll give identical choices and identical choice correlations. However, perfect redundancy never happens and there will always be some distinction between the neuronal noise that favors one readout pattern. In this case there is no freedom to alter the readout weights, so there is no meaningful way to describe the consequences of changing weights for optimal decoding. That is only meaningful for suboptimal decoding, and there are then too many possibilities to make any useful general claims about them here. As we now say in Section 3.2:

“More generally, inferring arbitrary suboptimal nonlinear decoders from choice correlations will be hard, and will require some knowledge of the joint response statistics [46,61].”

What role does the sparseness/broadness of tuning for different (relevant/nuisance) stimulus dimensions (or the rate modulation due to different stimulus features) in a population play?

Unfortunately there is no simple answer to this very interesting question. The effects will depend on the dimensionality of the target and nuisance variables, on the type of task, and especially on any interactions in the joint representation between the task-relevant and nuisance variables. Past work described the interplay between tuning and population size (Zhang and Sejnowski 1999), tuning and task type (Butts and Goldman 2006), tuning and dimensionality (Gao and Ganguli 2015). All of these considerations are important when assessing the role of sparsity in population information and choice correlation. An adequate answer to this question would require an entire paper, so we propose to not address it in the present submission.

What are the constraints that bound these measures?

This is now addressed when the NACCC is introduced. This measure of choice correlations is bounded between -1 and $+1$: as Eq 17 shows, the NACCC is expressible as a correlation coefficient based on the average conditional covariance matrix $\bar{\Gamma}$ (Section 4.2):

“ B_{Rk} is actually a correlation coefficient based on the average conditional covariance $\bar{\Gamma}$, and is bounded in absolute value by 1.”

Are these measures biased in some way at low trial numbers or with a response/choice bias towards one option in a 2AFC or a range of options in a continuous discrimination?

We focus here on unbiased choices, as it is most common to train animals to be unbiased, and properly treating a biased case introduces complexity that does not provide much additional conceptual insight (Haefner 2015).

A new **Figure S6** now shows the bias in estimating the decoding efficiency from choice correlations, as a function of the number of trials. When there are too few trials, this leads to an overestimation of the slope and therefore of the decoding efficiency. When these simulations use more than around 300 trials, this bias disappears.

Additionally, the tools presented in this manuscript seem like they would allow for important quantitative statements about the experimental demands for their use compared to standard linear approaches. What are the experimental contexts in which the author's approach buys an experimenter additional sensitivity? Can the authors outline the tradeoffs in terms of task design (eg 2AFC vs continuous report) or stimulus set (eg the number of relevant or irrelevant stimulus dimensions that change) that best make use of their approach. For example, one may assume that a continuous discrimination task will provide additional sensitivity for understanding the mapping between visual representations and choice and that this would be even more true with this decoding approach. But, is there a cost in terms of the amount of data (#s of neurons or # of trials) needed?

When the stimulus range is wider than the noise (signal-to-noise (SNR) > 1), continuous estimation tasks can help trace out a more complex stimulus-estimate relationship. The behavioral SNR defines an effective resolution for this mapping at which the neural encoding and decoding can be fruitfully explored. However, chains of binary discrimination can provide similar information about continuous estimates (Graf et al. 2011). For fine tasks, where SNR ~ 1, we do not see much difference between continuous estimation and binary discrimination. We now perform simulations to test this with redundant cubic codes, and report them in the new **Figure S6**.

For the tradeoff in terms of stimulus set (e.g. the number of relevant and nuisance variables), we already have a section describing what happens when experiments differ from natural environment (Discussion 3.3), saying: “we *should* see some nonlinear choice correlations even when nuisance variables are fixed. This is because neural circuitry must combine responses nonlinearly to eliminate natural nuisance variation, and any internal noise passing through those same channels will thereby influence the choice. In other words, nonlinear noise correlations need not predict a fixed stimulus, but they may predict the choice.” Thus, even in experiments where they don’t have irrelevant nuisance variability, we still expect our general approach to be applicable. However, the prediction for optimal choice correlations should only hold under conditions (including nuisance variation) in which a decoder was trained.

2. The authors spend a fair amount of time parsing the distinction between intrinsic and extrinsic noise and the subdivision of nuisance variability. This is important, but becomes muddled by the time the data in the final figure are addressed. The motivation from the start of the paper was to draw the distinction between properties of a stimulus that change firing rates (or patterns) but are not useful for guiding a discrimination. Unfortunately, these data don’t contain visual stimuli that map perfectly onto the examples that the authors developed earlier in the text and there is not a strictly irrelevant stimulus feature (like, say, phase). The authors treat the direction of the visual stimulus in the discrimination task as a nuisance variable, even though the direction of the stimulus itself is what the subject uses to determine the distribution that it originated from.

Ultimately, I do not fault the authors for this, but I think a little more care in the writing could help the reader through these issues.

In the orientation variance discrimination task, we agree that orientation is not a nuisance variable that is completely additively independent from the relevant category, as it was for spatial phase. Nonetheless, it can be viewed as an observable multiplicative combination of independent latent task-relevant and -irrelevant variables: observable orientation is $\phi = v s^{1/2}$ where $v \sim N(0, 1)$.

We now clarify this in the description of the task:

“The categorical target variable s is therefore the variance of the orientation distribution. This coarse binary discrimination task is a simplified version of a task that might arise in nature when identifying a surface texture or material [58]; the orientation of the material would be a nuisance variable independent of the material type. Here the observable variable, the orientation, is the product of the target variable and a nuisance variable v . An additional

nuisance variable was the stimulus contrast varying independently of the stimulus variance, although here we only analyze the highest contrast.”

Relatedly, the authors state:

“Combining the conclusions from these controls, we find no evidence that the brain optimally decodes any stimulus-dependent internal noise correlations in this task.”

Have the authors demonstrated that there actually are any stimulus-dependent internal noise correlations in this task? Do the authors simply mean that the distributions have changed between the shuffle analyses in E, F and G of Figure 6? Can they please clarify?

In this task, there appears to be little stimulus-dependent internal noise to decode. Figure 6F shuffles the internal noise while preserving the external nuisance variable, and the structure is preserved; Figure 6G shuffles the nuisance variable and the structure is abolished.

However, we directly examined the internal noise correlations too. We selected trials with nearly the same orientation, to fix the stimulus *and* nuisance variables, and then measured if there are stimulus-dependent internal noise covariances within these trials. Here, we plot joint responses for some example pairs of neurons. Some do show a weak orientation-dependent internal noise correlation, but this effect is much weaker than the shifts in the mean with orientation (a nuisance variable).

Figure S5: Internal noise covariance is only weakly tuned to orientation and insignificantly tuned to choice. **A:** Scatter plots of neural responses to multiple groups of trials, each group with nearly identical orientations ($\pm 1.5^\circ$). There are significant shifts in the means and variances of these response clouds, but changes in the correlations are not reliable. Cell pairs with strongest joint quadratic tuning were selected. **B:** For these same selected neurons, plots of linear and quadratic tunings confirm that the mean response is strongly tuned to orientation, the response variance is moderately tuned, and the internal noise cross-covariance is not tuned. **C:** Histograms of choice correlations show that purely internal noise is not significantly correlated with choices ($p < 0.01$, two-sample Kolmogorov-Smirnov test for a choice-shuffled null distribution), unlike the nuisance-generated fluctuations seen in Figure 6D. To isolate the correlation of internal noise on choice, we compute the Normalized Average Conditional Choice Correlation (NACCC) where we condition on, and then average over, the *complete* stimulus (s, v) rather than just on the task-relevant stimulus s as in Eq. 17. Individual choice correlations within the histograms are each colored by their significance according to their own null distribution (Methods 4.4).

Thus we see that there might be some weakly stimulus-dependent internal noise correlations providing nonlinear information. However, as we explained in Figure S2, any nonlinear information arising *without* nuisance variation need not be simply related to choice (Eq 127). Indeed, the nonlinear choice correlations for internal noise are not significant (two-sample Kolmogorov-Smirnov test). We now mention this in Section 2.3.2:

“We looked directly for stimulus-dependent internal noise correlations by conditioning on both the signal and the nuisance variable (which here is simply the single number, orientation) and measuring orientation-dependent response covariances. The resultant nonlinear tuning was quite weak compared with the trial-to-trial variability in those nonlinear statistics, and available nonlinear information arose largely in changing variances rather than covariances; likely arising from Poisson statistics and tuning of the mean firing (Supplementary Figure S5A,B). Internal noise fluctuations in those directions were not significantly correlated with choice (Supplementary Figure S5C, $p=0.088$, 0.830 , 0.969 for linear, square, and cross terms for monkey 1; $p=0.073$, 0.094 , 0.573 for linear, square, and cross terms for monkey 2 using a two-sample Kolmogorov-Smirnov test).”

3. Finally, this topic may be well beyond the scope of the current manuscript, but it will immediately spring to a reader’s mind. In the discussion, the authors note the potential limitations of this approach for addressing complex IT responses. But what about intermediate steps in visual processing? I’m curious to know whether the authors see this approach as being useful for understanding the transformation and construction of distinct visual features from subordinate features and how those responses are used to guide behavior.

Absolutely, this can be done, but it’s more complicated. We are actively working on generalizations for what we term recoding — the transformation of neural representations over time or across brain areas. A nice feature of relating neural responses to stimuli and choices, as we do in this paper, is that the relevant input and output quantities are naturally low-dimensional. In contrast, recoding is a mapping between high-dimensional neural spaces. Communication subspaces (Semedo et al 2019) provide a useful *linear* way of identifying low-dimensional interactions across brain areas, and we plan to generalize this approach to allow nonlinearities. Others have examined the type of nonlinear transformation that obtains between brain areas (Pagan, Simoncelli, Rust 2014). Our method may provide benefits beyond these past works by using a nonlinear targeted dimensionality reduction, where the selected dimensions focus on the overlap of the subspaces formed by both the stimulus-tuned and choice-tuned responses.

The concept of nuisance correlations seems to make more sense for an animal engaged in a task than it does for neurons within any particular visual brain area. Further, what is/is not a

nuisance may differ across (or even within) visual areas and one neuron's nuisance variable may well be another neuron's signal.

Nuisance variables are indeed specific to a task, and so nuisance *correlations* will also be specific to a task. In general we think it muddles concepts to consider nuisance variables to differ across neurons instead. Perhaps in very clear-cut cases it may be a reasonable shorthand to assume that some neurons are dedicated to an exclusive set of tasks. For example, retinal neurons are not used for auditory tasks. But even there, natural stimulus/task sets may have nontrivial multisensory associations; visual neurons might help localize and ignore an auditory distractor, which would then help isolate an auditory target. In statistics, there is an important concept of ancillary statistics, namely statistics which are not tuned to the parameter of interest. Importantly, ancillary statistics *can* help extract information from other tuned statistics (Basu 1959), and this situation applies to this audiovisual example. Zylberberg (2017) has a *BioRxiv* paper showing essentially how ancillary statistics can provide information for a specific model (though he doesn't call them ancillary statistics), but other than that, informative ancillary statistics have been essentially ignored by neuroscience, at least to our knowledge. This example emphasizes the danger of assuming that neurons contain dedicated information about certain task variables. Thus we advocate for keeping nuisance designations as task-specific, not neuron-specific.

For example, the authors use phase as an example in the text and in Figure 2. Similarly, they could have used spatial frequency, which is useful for the following, more complicated, example. If these Gabors drifted, their SF, drift rate (phase velocity) and orientation would be used to construct an estimate of the stimulus velocity. These features tend to be represented somewhat separably in V1 simple cells (eg, see Mazer et al, 2002; Priebe et al 2006) but are combined by complex cells in V1 or MT neurons to represent stimulus speed.

For this particular speed example, our formalism holds quite nicely: simple cells would encode information about velocity in their nonlinear statistics (e.g. second-order products, like for a Reichardt detector), whereas complex cells would have already performed these nonlinear computations and encode information directly in their linear statistics (firing rates). These two classes of neurons would be highly redundant because the complex cells' information is constructed from simple cells.

For a task to estimate object (group) velocity, phase and spatial frequency would be nuisance variables wherever they are encoded, and would induce nuisance correlations in V1 that would be attenuated to small variance in MT by averaging over V1 neurons. Conversely, when estimating spatial frequency, a stimulus that contains a variety of objects with different speeds would induce nuisance correlations in both V1 and MT. The identification as nuisance variables does not vary across areas, but the magnitude of nuisance correlations does vary.

Does this slight elaboration break down the tools developed here (as the authors suggest might happen in IT), or are they robust to this example? Could the magnitude of choice correlations in different populations of neurons provide insight into how a choice is formed

based on a 'constructed' visual property like speed? Does this example pose a problem for the current definition of 'nuisance' variability that the authors have employed?

The difficulty in using our approach in high-level areas like inferotemporal cortex is *complexity*. As we said in the text: "In complex tasks, like recognizing objects from images with many nuisance variables, most of the relevant information lives in higher-order statistics, and therefore requires more complex nonlinearities to extract. In such high-dimensional cases, our proposed test is unlikely to be useful." For this reason, we expect that it will be impractical to track fluctuations from retinal inputs through the multistage nonlinear processing from retina to IT. Nonetheless, it may be possible to derive some interesting properties using local expansions near a reference, much like what is done with classification images (Ahumada 1996).

However, as described above, we do not see a problem for our theory in addressing low-dimensional constructed task variables like speed of simple objects. Any input variations that are distinct from the constructed target variable serve as nuisance variables, and would induce nuisance correlations in neurons that have not yet built invariances to those variations.

Choice correlations in this example could indeed provide insight into the choice process. Under optimal decoding, if some statistic is informative about object speed, then it should correlate with behavioral reports of object speed. The information content is determined by the tuning of the statistic, where the tuning is based on averages over all trials with a fixed task-relevant variable, including averages over nuisance variables. Nuisance variation of spatial frequency and phase likely means that simple cells in V1 would be only weakly mean-tuned. In contrast, the covariances in V1 should be more strongly tuned to speed, which is why MT's squared pooling of V1 provides information. Optimal decoding thus predicts that V1 responses should be weakly correlated with behavioral reports of object speed, whereas quadratic functions of V1 activity should be more correlated with them. Both of these correlations should be computed as always by averaging over nuisance variables.

We are excited about the possibility of applying our test more broadly to gain insights into other computations, like this speed example suggested by the reviewer. Stay tuned!

REVIEWERS' COMMENTS

Reviewer #1 (Remarks to the Author):

The authors have adequately addressed my concerns. This is an unusual paper, as clearly evinced by each reviewer in our own ways. However, I believe it meets the standards of being both interesting and correct.

Minor grammatical point: the meaning of the addition "we found similar decoding efficiencies for two monkeys reported in Section 2.3" is ambiguous. I would suggest "we found similar decoding efficiencies for two monkeys AS WAS reported in Section 2.3"

Reviewer #2 (Remarks to the Author):

I find some of the technical clarifications helpful, and the additions to the text mostly positive. Nonetheless, I still feel that the core conceptual issues with respect to the limitations of the encoding-decoding framework as a whole, and to which degree the addition of nonlinearities qualitatively changes the picture have not been adequately addressed. I found the answers to these questions not particularly convincing, and the problem is amplified by the fact that the reply misunderstood some of the key points I was trying to make. The technical content is alright for what it is, and I am unlikely to change my mind about the overall utility of the approach so an additional round of review would not resolve the remaining issues.

Reviewer #3 (Remarks to the Author):

The authors have done a commendable job replying to the reviewer's comments. As a result, I believe that the manuscript has been improved in clarity and content.

The authors have adequately addressed the questions that I raised during the first round of reviews. All of my remaining comments are beyond the scope of the manuscript as it is currently presented.

The topic of this manuscript is timely and I expect that it will be well read and frequently cited.

REVIEWER COMMENTS

Reviewer #1 (Remarks to the Author):

The authors have adequately addressed my concerns. This is an unusual paper, as clearly evinced by each reviewer in our own ways. However, I believe it meets the standards of being both interesting and correct.

Minor grammatical point: the meaning of the addition “we found similar decoding efficiencies for two monkeys reported in Section 2.3” is ambiguous. I would suggest “we found similar decoding efficiencies for two monkeys AS WAS reported in Section 2.3”

We thank the reviewer for the appreciative comments. We have corrected this grammatical mistake.

Reviewer #2 (Remarks to the Author):

I find some of the technical clarifications helpful, and the additions to the text mostly positive. Nonetheless, I still feel that the core conceptual issues with respect to the limitations of the encoding-decoding framework as a whole, and to which degree the addition of nonlinearities qualitatively changes the picture have not been adequately addressed. I found the answers to these questions not particularly convincing, and the problem is amplified by the fact that the reply misunderstood some of the key points I was trying to make. The technical content is alright for what it is, and I am unlikely to change my mind about the overall utility of the approach so an additional round of review would not resolve the remaining issues.

We are sorry that the reviewer feels that our reply misunderstood his key points. Perhaps in the future, the reviewer may wish to discuss these points in person (without reference to the review process and without breaking anonymity).

Reviewer #3 (Remarks to the Author):

The authors have done a commendable job replying to the reviewer's comments. As a result, I believe that the manuscript has been improved in clarity and content.

The authors have adequately addressed the questions that I raised during the first round of reviews. All of my remaining comments are beyond the scope of the manuscript as it is currently presented.

The topic of this manuscript is timely and I expect that it will be well read and frequently cited.

We thank the reviewer for the appreciative comments.